# High-resolution spatial distribution maps of road transport exhaust emissions in Chile, 1990 – 2020.

Mauricio Osses[1,3], Néstor Rojas[2], Cecilia Ibarra[3,4], Víctor Valdebenito[1], Ignacio Laengle[1], Nicolás Pantoja[1,3], Darío Osses[4], Kevin Basoa[3], Sebastián Tolvett[5], Nicolás Huneeus[3,4], Laura Gallardo[3,4], Benjamín Gómez[1,3]

[1] Universidad Técnica Federico Santa María (UTFSM), Santiago, Chile

[2] Universidad Nacional de Colombia, Bogotá (UNAL), Colombia

[3] Center for Climate and Resilience Research (CR)2, Santiago, Chile

[4] Universidad de Chile, Santiago, Chile

[5] Universidad Tecnológica Metropolitana, Santiago, Chile

*Correspondence to*: Mauricio Osses (mauricio.osses@usm.cl)

## Abstract.

This description paper presents a detailed and consistent estimate and analysis of exhaust pollutant emissions generated by Chile´s road transport activity for the period 1990-2020. The complete database for the period 1990-2020 is available at doi: http://dx.doi.org/10.17632/z69m8xm843.2 (Osses et al., 2021). Emissions are provided at high-spatial resolution (0.01º x 0.01º) over continental Chile from 18.5 S to 53.2 S, including local pollutants (CO, VOC, NOx, $PM_{2.5}$), black carbon (BC) and greenhouse gases ($CO_2$, $CH_4$). The methodology considers 70 vehicle types, based on ten vehicle categories, subdivided into two fuel types and seven emission standards. Vehicle activity was calculated based on official databases of vehicle records and vehicle flow counts. Fuel consumption was calculated based on vehicle activity and contrasted with fuel sales, to calibrate the initial dataset. Emission factors come mainly from COPERT 5, adapted to local conditions in the 15 political regions of Chile, based on emission standards and fuel quality. While vehicle fleet has grown fivefold between 1990 and 2020, $CO_2$ emissions have followed this trend at a lower rate and emissions of air local pollutants have decreased, due to stricter abatement technologies, better fuel quality and enforcement of emission standards. In other words, there has been decoupling between fleet growth and emissions' rate of change. Results were contrasted with global datasets (EDGAR, CAMS, CEDS), showing similarities in $CO_2$ estimations and striking differences in PM, BC and CO; in the case of NOx and $CH_4$ there is coincidence only until 2008. In all cases of divergent results, global datasets estimate higher emissions.

## 1 Introduction

Building and updating emission inventories provides key information for designing and evaluating public policies concerning topics relevant for the inhabitants of cities' quality of life, the environment and for mitigation of climate change (Kuenen et al., 2014; Creutzig et al, 2015). In international and national experiences, the construction of reliable emission inventories for road transport has been a bottleneck in mapping the emissions in cities (Zheng et al., 2014). In general, these difficulties occur due to two main reasons: the lack of disaggregated data at a level to construct detailed inventories, and the many variables to

consider when modelling emissions, increasing the uncertainty in the estimation of total emissions (Bond et al., 2004; Tolvett et al., 2016).

Latin America has an urbanization rate of more than 80% and cities suffer changes in local climate and air pollution, imposing big challenges to cope with (Henríquez and Romero, 2019; Hardoy and Romero-Lankao, 2011). Transport in cities is a major

source of air pollution and emissions of greenhouse gases (GHG) (Huneeus et al 2020a). Reliable inventories are needed to assess policy measures for air quality and climate change. In the case of Santiago, Chile's capital, there are good examples of the use of local data to analyse the impact of emissions on air pollution (Mazzeo et al., 2018), health benefits of policy scenarios (Mena-Carrasco et al., 2012), and for retrospective evaluation of the evolution of mobility and air quality, relating them to policy measures (Gallardo et al., 2018).


Despite research results for specific data analysis, inventories for Chile have huge scope for improvement in terms of data accuracy and disaggregation. Better inventories are needed for decision making related to cities' air quality and for climate change commitments. Barraza et al (2017) estimated that the motor vehicles were responsible for 37.3 % of $PM_{2.5}$ in Santiago, which highlight the role played by mobile sources in large urban areas. The transport sector at large accounts for 25% of 2018-

$CO_2$ estimates in Chile (MMA, 2021).  Further, in 2016, a fraction of 7 % of total BC is linked to the on-road transportation sector (Gallardo et al, 2020). International inventories include data for Chile, for example, EDGAR V4.3.2 covers between 1970 and 2012, later until 2015 (Crippa et al., 2019) and recently EDGAR V5.0 extended its range until 2018 (Crippa et al., 2021; Crippa et al, 2020).

Chile has two separated inventories, one for GHG and another for criteria pollutants of air quality. The GHG inventory follows the methodology established by the Intergovernmental Panel for Climate Change, it has been systematically kept since 2012, and it includes estimates starting in 1990 (MMA, 2019). This inventory is made in house at the Ministry of the Environment in coordination with other sectorial ministries, which assures capacity building within the ministry. Although this data set is well evaluated and consistent, the level of aggregation is national or at best at split into political regions. On the other hand,

inventories for criteria pollutants are built in connection with the establishment of attainment plans and produced through external consultancies, which has resulted in a scattered picture regarding reference years, emission factors used, and cities

along continental Chile. These inventories only consider specific industrial complexes or urban areas but not rural areas or background conditions, which limits the scope of attainment plans as highlighted by Huneeus et al (2020b).

The recent commitment made by Chile before the Paris Agreement and the United Nations Framework Convention on Climate Change (UNFCCC), considers achieving carbon neutrality regarding GHG by 2050, and reducing black carbon emissions by at least 25% by 2030 respect to levels in 2016 (Gobierno de Chile, 2020). To accurately monitoring progress with respect to BC emissions will require an improved spatial resolution and explicit monitoring of BC in $PM_{2.5}$ (Gallardo et al, 2020). The same applies when considering health impacts linked to PM and BC (Burnett et al, 2018, Kirrane, 2019)


Chile includes short lived climate pollutants (SLCP) in its inventories since 2012, when the Ministry of the Environment joined the Climate and Clean Air Coalition (CCAC) and committed to reduce emissions of SLCP. One of the actions taken was to identify the main sources of these pollutants, concluding that transport was a main emission sector and that black carbon concentrations were worryingly high (Jorquera et al., 2017). Data on black carbon emissions was recently updated showing

the situation has not changed significantly (Gallardo et al, 2020). Black carbon (BC) is a pollutant with impact on human health and contributes to climate change in a global and local scale (Bond et al., 2013; Hadley and Kirchstetter, 2012; Ramanathan and Carmichael, 2008), hence motivating the inclusion of a goal for black carbon reduction in Chile´s National Determined Contributions to the CMNUCC (Gobierno de Chile, 2020).

This description paper presents an extension and update of the data for the emissions inventory for on-road transport in Chile, considering fuel consumption records, vehicle fleet, stricter emission standards requirements and inclusion of motorcycles as a vehicular category. This inventory is based on previous methodologies (MAPS, 2013; SECTRA; Jorquera et al, 2017; Gallardo et al., 2020; Zheng, 2014). It incorporates the latest data available, with the purpose of calculating transport activity to obtain an updated estimate of transport emissions, offering high-resolution spatially distributed maps for Chile.


The paper is structured as follows: Section 2 describes the methodology; Section 3 presents main results showing the evolution vehicle technologies and their impact on emissions, and regional differences; Section 4 makes a comparison of results with EDGAR, CAMS, CEDS datasets for the period 1990 to 2015; Section 5 concludes.

**2 Methodology and data**

The annual emissions database provides estimates of exhaust emissions for on-road transport, i.e., vehicles traveling on public routes, nationwide, in urban and rural areas, for years 1990 to 2020. It does not include rail, air and sea transport modes and off-road machinery. The calculation of emissions was based on estimations of the number of vehicles and their activity level

by political region, which were used to calculate fuel consumption by vehicle category and, subsequently, exhaust emissions, as summarized in the methodological diagram (Fig. 1) and explained in this section. The resultant emission database was spatially distributed at 0.01x0.01 degrees of latitude and longitude resolution.

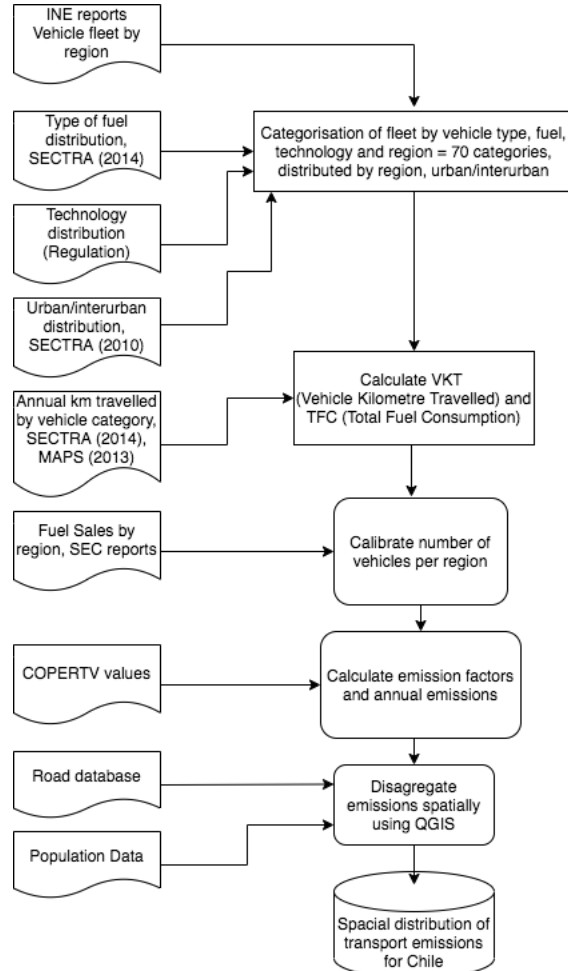

Figure 1. Methodological diagram

## 2.1 Vehicle fleet composition

Vehicle fleet composition was based on official government data on annual registration of in-use vehicles, i.e., the vehicles that each year pay their circulation permit after having approved the periodic technical inspection. The National Institute of Statistics (INE, https://www.ine.cl/) provides annual reports of the total number of vehicles with circulation permit per political region. Vehicle categories reported are light passenger, commercial and taxi vehicles; 12- and 18-meter buses; light, medium and heavy-duty trucks; and two-wheeled motorized vehicles. Since the emission factors and fuel consumption of the distinct

types of vehicles are being modelled using the Computer Programme to calculate Emissions from Road Transport (COPERT) version 5 developed by the European Consortium of Transport Impact (EMISIA, https://www.emisia.com) (Ntziachristos et al., 2009), the equivalence between Chilean INE categories and COPERT is shown in Table 1.

Table 1. Equivalence of European and Chilean main vehicle categories

| Europe COPERT | Chile INE | Category Code |
|---|---|---|
| Passenger cars (diesel and gasoline, $M_1$) | Automobile, station wagon, jeep, taxi | LPV |
| Light commercial vehicles (diesel and gasoline, $N_1$) | Van, pickup truck, minibus | LCV |
| Urban Bus Standard, Urban Bus Midi | Bus, taxi bus | UBS |
| Urban Articulated Bus | Bus | UBA |
| Intercity bus (Coach Standard) | Bus | ICB |
| HDT (Rigid <=14 t) | Truck | LDT |
| HDT (Rigid 14 t - 28 t) | Truck | MDT |
| HDT (Rigid > 28 t and articulated) | Truck, trailer truck | HDT |
| L-Category (Motorcycles) | Motorcycles and similar | MOT |

Each of these categories was subdivided to distinguish the type of fuel used (gasoline or diesel) using the most recent information from the Ministry of Transport and Telecommunications (MTT) and the secretariat for transportation (SECTRA, 2014) and the technology used, based on the emission standard in its European equivalent (EURO standard). To model the

vehicle technology distribution, we used information given in the Supreme Decrees of MTT and the Ministry of the Environment (MMA), corresponding to the enforcement of emission technologies standards for new vehicles entering the national fleet (all vehicles are imported), for the distinct vehicular categories distinguishing between regions.

Table 2. Introduction of emission standards for vehicle categories in Chile

| Category | Region | Year of exigency | | | | | | | | | | | | | |
|---|---|---|---|---|---|---|---|---|---|---|---|---|---|---|---|
| | | 1992 | 1993 | 1994 | 1996 | 1998 | 2002 | 2005 | 2006 | 2011 | 2012 | 2013 | 2014 | 2018 | 2020 |
| Gasoline LPV | XIII | EPA 83 | | | | Euro 1 | | Euro 3 | | Euro 4 | | | Euro 5 | | Euro 6 |
| | V and VI | EPA 83 | | | | | | Euro 1 | Euro 3 | | | Euro 4 | Euro 5 | | |
| | Rest of the country | | | EPA 83 | | | | Euro 1 | Euro 3 | | | Euro 4 | Euro 5 | | |
| Diesel LCV | XIII | EPA 83 | | Euro 1 | | Euro 1 | | Euro 3 | Euro 4 | Euro V | | | | | |
| | V and VI | EPA 83 | | | | | | Euro 1 | Euro 3 | | | Euro 4 | Euro 5 | | |
| | Rest of the country | | EPA 83 | | | | | Euro 1 | Euro 3 | | | Euro 4 | Euro 5 | | |

| Category | Region | | | | | | | | |
|---|---|---|---|---|---|---|---|---|---|
| Trucks LDT MDT HDT | XIII | Euro I | Euro II | | Euro III | Euro III DPF | Euro IV | Euro V | |
| | IV to X | | | | Euro III | | Euro IV | Euro V | |
| | II and III | | | | Euro II | | Euro IV | Euro V | |
| | Rest of the country | | | | Euro II | | | | |
| Buses UBS UBA ICB | XIII public transport | Euro I | Euro II | Euro III | | Euro III DPF | Euro IV DPF | Euro V | Euro VI |
| | XIII other buses | Euro I | Euro II | | Euro III | | Euro IV | Euro V | |
| | IV to X | Euro I | Euro II | | Euro III | | Euro IV | Euro V | |
| | Rest of the country | | | | Euro II | | Euro IV | Euro V | |
| 2-wheeler MOT | All regions | Euro 1 | | | Euro 2 | Euro 3 | | | |

Information in Table 1 is based on the following government decrees: DS82/93MTT, DS 54/94MTT, DS 55/94MTT, DS 130/2002MMT, DS4/2012MMA, available at https://www.bcn.cl/leychile/.

The combination of categories, fuels and emission standards generates a total of 70 types of vehicles for the emission analysis, distributed over political regions and distinguishing between urban and interurban activity. The distribution of vehicles into

urban and interurban activity per region was based on a proportional regional distribution according to SECTRA (2010).

## 2.2 Calculation of vehicle activity and fuel consumption

Activity level is expressed as *VKT* (vehicle kilometre travelled) calculated as the sum of the vehicles in each type per kilometres driven (Eq. 1).

$$VKT = \sum_{i,j,k}^{N} N_{i,j,k} \cdot KM_{i,j,k}$$

(1)

where $N_{i,j,k}$ is the number of vehicles of type i in region j and road class k (urban or interurban); $KM_{i,j,k}$ are the kilometres travelled per year by vehicles type i, in region j and road class k.

The kilometres travelled by each type of vehicle used in equation 1 are shown in Table 3. They correspond to estimations by SECTRA (2014) and MAPS (2013) for the first level of vehicle aggregation (main vehicle categories on Table 1).


Table 3. Annual activity level per region and vehicle type (*AL*). Currently, there are 15 political regions in Chile numbered from I to XV, being XIII the Metropolitan Region of Santiago.

| | Year kilometres travelled [km yr⁻¹] | | | | | |
|---|---|---|---|---|---|---|
| Region | LPV | LCV | MOT | Taxi | Bus | Truck |
| I, XIV | 8,228 | 9,873 | 5,000 | 16,455 | 40,361 | 30,271 |
| II | 13,302 | 15,962 | 5,000 | 26,604 | 66,618 | 46,120 |
| III | 14,382 | 17,259 | 5,000 | 28,765 | 57,942 | 43,456 |
| IV | 15,241 | 18,289 | 5,000 | 30,482 | 54,292 | 37,587 |
| V | 13,986 | 16,784 | 5,000 | 27,973 | 33,598 | 23,260 |
| VI | 12,127 | 14,552 | 5,000 | 24,254 | 28,935 | 20,032 |
| VII | 12,582 | 15,099 | 5,000 | 25,165 | 39,394 | 27,273 |
| VIII | 12,390 | 14,869 | 5,000 | 24,781 | 54,455 | 37,700 |
| IX | 13,515 | 16,217 | 5,000 | 27,029 | 56,173 | 38,889 |
| X, XV | 14,494 | 17,393 | 5,000 | 28,989 | 58,059 | 40,195 |
| XI | 12,089 | 14,507 | 5,000 | 24,178 | 31,175 | 21,175 |
| XII | 7,284 | 8,741 | 5,000 | 14,569 | 45,503 | 34,877 |
| XIII | 14,956 | 17,948 | 5,000 | 37,405 | 67,368 | 29,471 |

Source: authors elaboration based on SECTRA (2014)


Once the number of vehicles per region has been obtained, the Total Fuel Consumption (TFC) for a given year was calculated as shown in Eq. (2):

$$TFC = \sum_{ijklm} \frac{AL_{ijkl} \times N_{ij} \times X_{1_{ijk}} \times X_{2_{ijl}} \times X_{3_{ijm}}}{FC_{ijklm}}$$

(2)

where $i$ is the region to which the vehicle belongs; $j$ represents the vehicle type (passenger, commercial, bus, truck, motorcycle); $k$ represents the share of the subcategory for the distinct vehicle types: for passenger and commercial vehicles the subcategories are diesel and gasoline, for buses are rigid, articulated and intercity and for trucks are light, medium and heavy duty; $l$ represents the share of vehicles that drive on urban areas or interurban roads; $m$ is the share of vehicles according to their emission control technology: Pre Euro I, Euro I – VI. The terms of Equation 2 represent the following: *AL* is the annual

activity level of the vehicle $[km.\,year^{-1}.\,veh^{-1}]$, $N$ is the total number of registered vehicles per year and per region where the sub-indices disaggregate this amount into the subcategories mentioned, $FC$ is the fuel consumption per vehicle $[km.\,lt^{-1}]$, $X_1$ is the percentage of the distinct subcategories, $X_2$ is the share of traffic counts depending if the vehicle is driven in urban or rural areas and $X_3$ is the percentage of vehicles with distinct emission control technologies. Thus, Equation 2 is used to calculate the total fuel consumption for each category presented as sub-indices.


TFC was compared to real fuel sales for each region. The Electricity and Fuel Superintendence (SEC, www.sec.cl) provides information on sales of diesel and gasoline for the transportation sector, by political region. The calculated TFC (see Equation 2) was compared to the data given by SEC and then a correction factor to the total number of registered vehicles in each region is applied to make these two fuel consumptions equal, correcting for those vehicles that are registered but do not contribute to actual driving activity. Thus, the number of active vehicles in a region was inferred and adjusted accordingly. A comparison between official figures of national fuel sales (SEC) and estimated TFC, for gasoline and diesel at a national level, is shown in Figure 2.

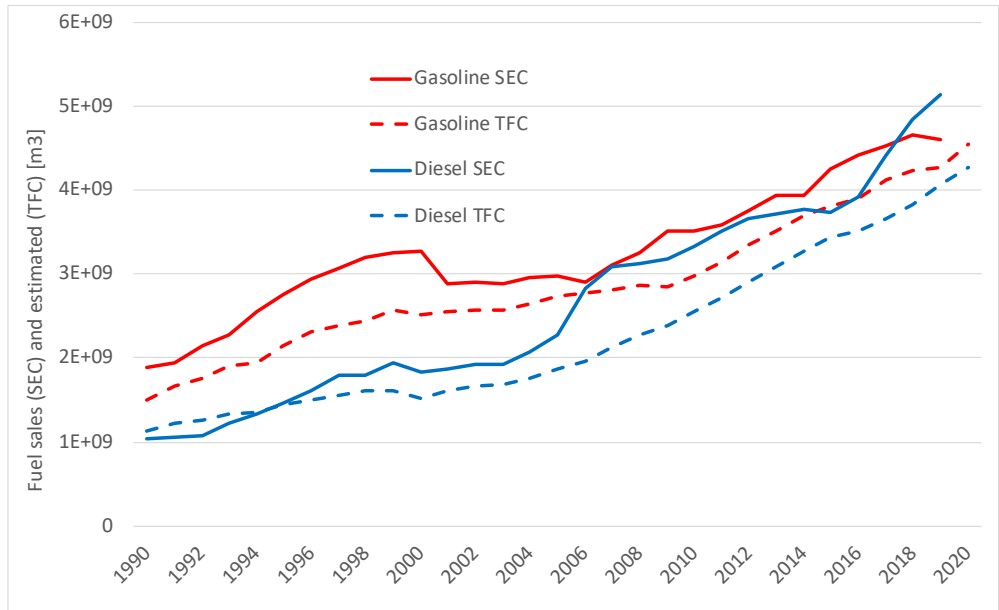

Figure 2. National fuel sales and estimated total fuel consumption for gasoline and diesel in Chile

In general, estimates of fuel consumption are lower than fuel sales, which means the number of registered vehicles generates lower activity than reality or specific consumption factors for each vehicle technology are lower than the real driving conditions in Chile. These differences are addressed increasing the number of vehicles according to their technology, matching official fuel sales figures.

### 2.3. Emission factors and annual emissions

The estimation of emissions considered that all vehicles that enter Chile are required to comply with the European EURO regulations or their US equivalent, according to the regulation requirements on Table 2 (Introduction of emission standards for vehicle categories in Chile). The assignment of emission factors for each of these vehicle types was carried out by applying COPERT 5 values (EEA, 2019), adapted to the Chilean fleet (Gómez, 2020). Total emissions are calculated multiplying *VKT*

by an emission factor in grams per kilometre. The result is a regional emission database distinguishing by urban and interurban emissions, for CO, $CO_2$, VOC, NOx, $PM_{2.5}$, $CH_4$ and BC.

With the calculated mobility demand, emissions of pollutants based on vehicle-kilometres travelled by the different vehicular categories can be estimated through Eq. (3) (Zheng et al., 2014):

$$EMIS_{n,m,l,k,j.i} = \sum_{ijklnm} VKT_{ijklm} \times EF_{ijklmn} \qquad (3)$$

where the sub-index n represents the pollutant: $CO_2$, CO, $NO_X$, $PM_{2.5}$, BC, VOC and $CH_4$. EF is the emission factor for the
pollutant n $[gr.km^{-1}]$ for each vehicle.

$CO_2$, CO, $NO_X$, $PM_{2.5}$, VOC emission factors are being modelled using COPERT 5 methodology (Ntziachristos et al., 2009) based on the average speed in the driving cycles given by previous studies in all regions of Chile (Osses et al., 2014). $CH_4$ emission factors used are from previous reports (Ntziachristos et al., 2007; USEPA, 2018). However, COPERT 5 does not
model black carbon emission factors due to the difficulty of the classification of this type of aerosol (Bond et al., 2013). Nonetheless, there have been studies to determine black carbon fractions in particulate matter, distinguishing between vehicle technology and fuel type, considering that elemental carbon and black carbon fractions are equivalent (Bond et al., 2004; Chow et al., 2010; Minjares et al., 2014; Ntziachristos et al., 2007). These fraction values can be used to obtain BC emission factors as follows:


$$EF_{BC} = EF_{PM} F_{2.5} F_{BC} \qquad (4)$$

where $EF_{PM}$ is the emission factor of the total particulate matter in the exhaust, $F_{2.5}$ is the mass fraction of particles that have an aerodynamic diameter of 2.5 $\mu m$ or less and $F_{BC}$ is the fraction of black carbon in these particles. The mass fraction of fine particles used is 0.9 since in a generic and ideal particle distribution, between 80 and 95% of the total mass of the particles are concentrated in this range (Payri and Desantes, 2011). Table 4 shows the black carbon fractions used, distinguishing vehicle
category, motorization and vehicle technology. Motorcycle BC emission factors used were taken from the literature (Cai et al., 2013).

Table 4. BC/PM$_{2.5}$ fractions for the vehicle emission technologies in Chile.

| Vehicle Category | Pre Euro [%] | Euro I/1 [%] | Euro II/2 [%] | Euro III/3 [%] | Euro IV/4 [%] | Euro V/5 [%] | Euro VI/6 [%] |
|---|---|---|---|---|---|---|---|
| Bus | 50 | 65 | 65 | 70 | 75 | 75 | 15 |
| Light duty truck | 55 | 70 | 80 | 85 | 87 | 10 | 29 |
| Medium duty truck | 47 | 70 | 81 | 72 | 69 | 23 | 25 |
| Heavy duty truck | 50 | 65 | 65 | 70 | 75 | 75 | 15 |
| Commercial and passenger light diesel vehicle | 47 | 70 | 80 | 72 | 69 | 25 | 25 |
| Commercial and passenger light gasoline vehicle | 30 | 25 | 25 | 25 | 15 | 17 | 17 |

Roman number notation applies for heavy duty vehicles and Arabic numbers apply to light commercial and passenger vehicles.

COPERT V considers correction of emission factors by vehicle age for light vehicle categories EURO 3 & 4 and for VOC, CO, NOx. These corrections were also applied.

**2.4 Spatial disaggregation**

The spatial distribution of transport emissions per political region consists of allocating Gg of emissions per year to each cell in a grid, with cells of 0.01x0.01 degrees of latitude and longitude covering the fifteen regions in the country. The regions correspond to the political administrative division of the territory. The distribution depends on the types of roads in each cell, the vehicle flow, and the presence of urban population.

The identification of roads in each cell was based on road network maps, available from the official roads database for Chile´s and the official regional limits (BCN, 2020). The information on Chile´s road network was complemented with data from OpenStreetMap (OSM, 2020). It covers a total of 77.800 km of rural and urban roads. Each road on the network was classified into a hierarchy comprising freeways, arterials, collectors, and local roads. The estimation of vehicle flow on each type of road

resulted from applying a road weight factor, based on toll barrier vehicle counts at interurban roads (MOP, 2020) and origin-destiny surveys in urban roads. Average weight factors are 54% for freeways, 23% on arterials, 16% for collectors and 7% on local roads. The road weight factors vary by region, urban and interurban areas, and cities in a region, and are provided by the Transport Secretariat, SECTRA (Osses et al., 2010).

Emissions were distributed over the road network using QGIS opensource software. Urban emissions were also distributed among the cities of each region according to their population (INE, 2017). QGIS allocates emissions to cells based on the type of roads, with their weight factor, and the presence of cities. Therefore, emissions for each cell depend on the roads and the

presence of urban population. The Transport Secretariat, SECTRA, provides the proportion of urban and interurban roads per region (Osses et al., 2010) and the urban areas of each region can be associated to cities with population over 5000 inhabitants.

The number of kilometres in each cell is proportional to the annual emission for each cell in the grid, and the sum of emissions in all cells coincide with the total emission assigned to each region of the country.

## 3 Results and discussion

### 3.1 Evolution of fuel quality, vehicle technology and emission factors

Based on the information given in sections 2.1 and 2.2, the number of vehicles and their technological evolution has been
determined, according to European emission standards (Pre EURO, EURO 1-6 for light duty vehicles, EURO I-VI for trucks and buses). The enforcement of stricter emission standards along the country has been sustained by permanent national fuel quality improvements. The reduction of sulphur in fuels have been progressive since 1990 to 2004 from 5000 ppm to 50 ppm of sulphur in diesel and from 1000 ppm to 30 ppm in gasoline. In April 2001, the elimination of leaded gasoline was made effective, which allowed national enforcement of three-way catalytic converters, reducing the levels of CO, VOC, NOx and
also particulate matter from motor vehicles (Moreno et al., 2010). Since 2004 the levels of ppm of sulphur in diesel and gasoline are regulated by law for the whole country. In the year 2010 the standard was fixed in 10 ppm for gasoline [DS N°66 MMA, 2010] and 15 ppm for diesel since 2012 (DS40.263/2012 MMA). Currently, with the introduction of EURO 6/VI emission standards (DS40/2019 MMA), the maximum limit of sulphur in diesel is set at 10 ppm in the country. These policies, involving the introduction of better fuel quality and stricter emission standards (Table 2), are expected to keep decreasing the exhaust
emissions of local pollutants from on-road transportation, even considering the permanent increase of mobility. The evolution of emission standards and number of vehicles for the whole country is shown in Figures 3 and 4, adding up specific regional information from 1990 until 2020.

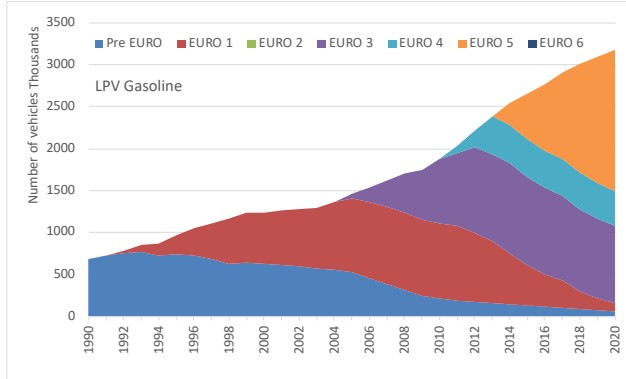
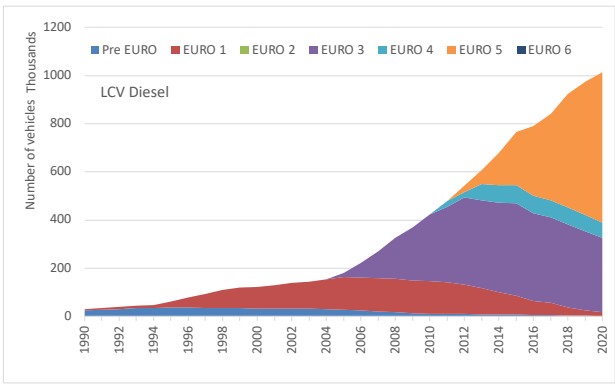

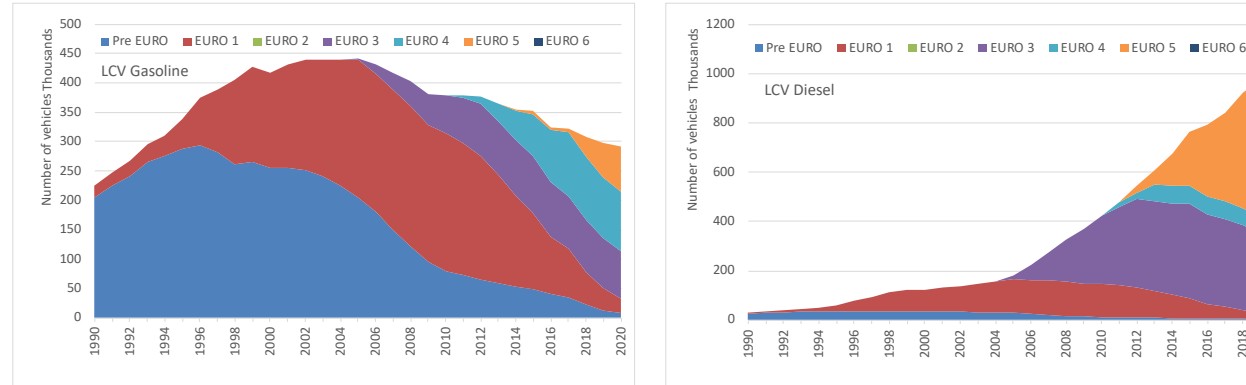

Figure 3. Emission standards and vehicle fleet in Chile, 1990-2020, for light passenger vehicles (LPV) and light commercial vehicles (LCV), using gasoline and diesel respectively

Figure 3 shows a continuous growth of light vehicles in Chile, except for the LCV-gasoline category, whose fall is offset by a strong increase in LCV-diesel vehicles. The total fleet of active light vehicles, both gasoline and diesel, grew from 968 thousand to 5.1 million units between 1990 and 2020. The gradual disappearance of technologies prior to EURO requirements (Pre EURO) and EURO 1 standards is also observed, ending in 2020 with a fleet mixed between EURO 3 and EURO 5. By 2020 there are already a few EURO 6 vehicles, but they cannot be distinguished in the graphs.

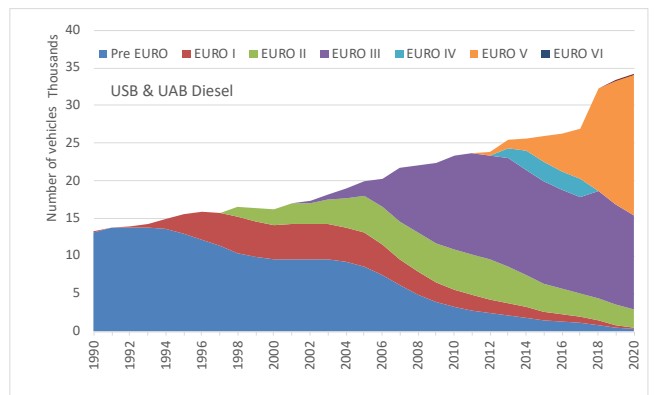

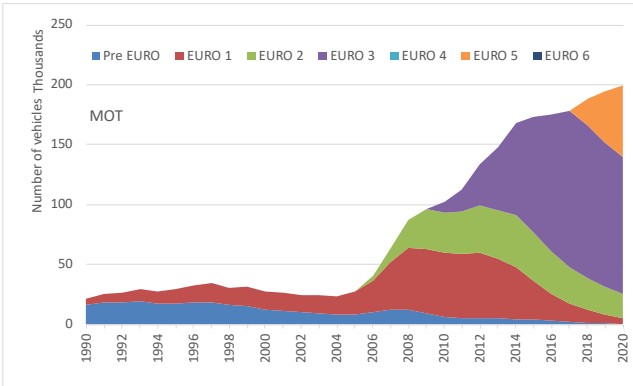

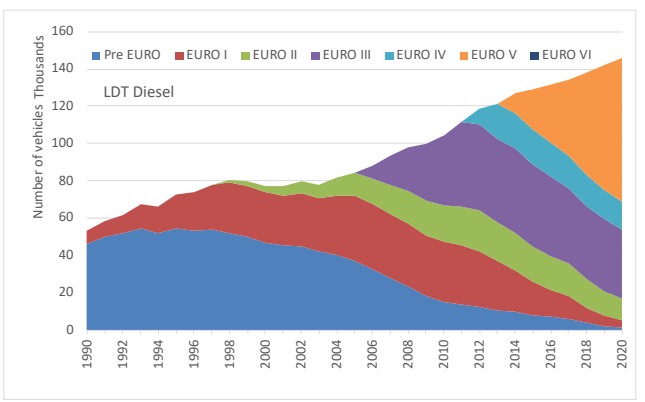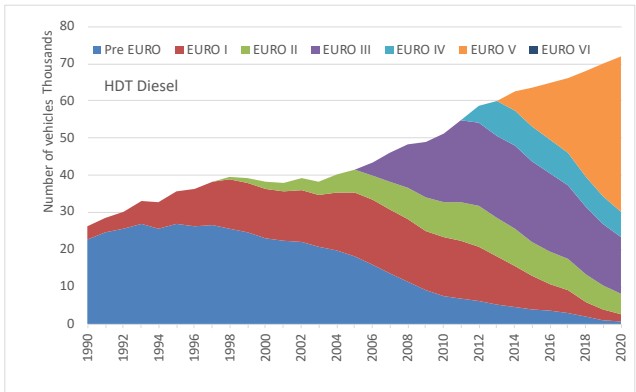

Figure 4. Emission standards and vehicle fleet in Chile, 1990-2020, for urban standard buses (USB), urban articulated buses (UAB), 2-wheelers (MOT), light-duty trucks (LDT) and heavy-duty trucks (HDT)

The vehicles in categories of heavy diesel vehicles have similar behaviors. The number of heavy diesel vehicle units shows a sustained growth, going from 92 thousand in 1990 to 250 thousand in 2020, adding all types of buses and trucks in Chile. The introduction of standards has been gradual, with the disappearance of Pre EURO and EURO I vehicles in 2020, leaving a mixed fleet between EURO II - EURO V at the end of the analysis period. By 2020, the first low-emission buses for public transport were operating in the Metropolitan Region, with a total of 700 EURO VI units and 400 electric buses, which is almost imperceptible in the upper left graph of Figure 4.

Two-wheel vehicles were not important in Chile until 2006, when it started to increase, all of them 4-stroke. The fleet of 22 thousand units in 1990 grew to 200 thousand motorcycles in 2020, most of them complying with the EURO 4 standard. A growing urban congestion and the proliferation of home delivery services can explain this higher demand for motorcycles in Chile.

Knowing the different vehicle technologies in Chile and their equivalence with the European categories (Table 1), it is possible to obtain the emission factors from the values reported by the COPERT model (Ntziachristos et al., 2009). For doing this, different activity parameters are considered, where the most relevant are the average speed of displacement and load level. Table 6 shows the result of this assignment, considering urban speeds for light vehicles, buses and motorcycles, and interurban speeds for trucks. The vehicle categories correspond to those indicated in Table 5.

Table 5. Emission factors used for different compounds and vehicle categories in Chile, the numbers 0-6 indicate the corresponding EURO standards, and the letters G/D correspond to gasoline or diesel respectively.

| Vehicle ID | CO$_2$ [g/km] | NOx [g/km] | PM$_{2.5}$ [g/km]x10$^3$ | BC [g/km]x10$^3$ | CO [g/km]x10$^2$ | VOC [g/km]x10$^2$ |
|---|---|---|---|---|---|---|
| LPV0G / LPV0D | 262,2 / 179,77 | 2,41 / 0,53 | 2,56 / 197,83 | 0,76 / 59,35 | 2569.67 / 60.41 | 217.42 / 13.4 |
| LPV1G / LPV1D | 186,02 / 155,99 | 0,33 / 0,64 | 2,56 / 72,31 | 0,76 / 21,69 | 186.82 / 37.52 | 16.96 / 4.77 |
| LPV3G / LPV3D | 189,48 / 155,06 | 0,06 / 0,74 | 1,23 / 34,21 | 0,37 / 10,26 | 69.02 / 8.02 | 1.85 / 1.65 |
| LPV4G / LPV4D | 195,82 / 155,06 | 0,04 / 0,56 | 1,23 / 28,59 | 0,37 / 8,57 | 28.37 / 7.58 | 1.33 / 1.08 |
| LPV5G / LPV5D | 195,82 / 155,06 | 0,02 / 0,59 | 1,42 / 2,24 | 0,42 / 0,67 | 29.57 / 3.92 | 0.64 / 0.09 |
| LPV6G / LPV6D | 195,82 / 155,06 | 0,02 / 0,48 | 1,5 / 1,57 | 0,45 / 0,47 | 26.34 / 5.25 | 0.67 / 0.09 |
| LCV0G / LCV0D | 254,6 / 263,84 | 2,95 / 1,69 | 2,56 / 304,63 | 0,76 / 91,38 | 1667.32 / 118.4 | 160.6 / 12.64 |
| LCV1G / LCV1D | 298,68 / 236,59 | 0,45 / 1,19 | 2,56 / 96,94 | 0,76 / 29,08 | 454.88 / 49.93 | 18.88 / 12.64 |
| LCV3G / LCV3D | 298,68 / 236,59 | 0,09 / 1 | 1,23 / 64,95 | 0,37 / 19,48 | 236.54 / 40.94 | 2.64 / 7.84 |
| LCV4G / LCV4D | 298,68 / 236,59 | 0,04 / 0,81 | 1,23 / 33,93 | 0,37 / 10,17 | 127.36 / 32.45 | 1.13 / 2.9 |
| LCV5G / LCV5D | 176,44 / 222,44 | 0,01 / 1,3 | 1,79 / 1,15 | 0,53 / 0,34 | 54.73 / 0.02 | 0.7 / 0.01 |
| LCV6G / LCV6D | 176,44 / 222,44 | 0,01 / 1,05 | 1,79 / 1,15 | 0,53 / 0,34 | 54.73 / 0.02 | 0.7 / 0.01 |
| LDT0 / HDT0 | 493,01 / 944,4 | 6,51 / 11,7 | 307,78 / 456,57 | 92,33 / 136,97 | 209.78 / 242.71 | 97.88 / 74.69 |
| LDT1 / HDT1 | 422,69 / 809,91 | 4,16 / 8,26 | 159,18 / 358,77 | 47,75 / 107,63 | 88.52 / 197.02 | 29.37 / 66.55 |
| LDT2 / HDT2 | 405,64 / 784,26 | 4,48 / 9,02 | 83,67 / 172,92 | 25,1 / 51,87 | 73.75 / 161.41 | 19.51 / 44.54 |
| LDT3 / HDT3 | 427,7 / 816,16 | 3,53 / 7,29 | 79,58 / 176,38 | 23,87 / 52,91 | 85.59 / 194.45 | 18.06 / 41.21 |
| LDT4 / HDT4 | 413,89 / 774,93 | 2,41 / 4,94 | 19,2 / 38,47 | 5,76 / 11,54 | 43.74 / 93.83 | 2.62 / 5.45 |
| LDT5 / HDT5 | 398,37 / 754,6 | 2,43 / 5,1 | 22,83 / 49,14 | 6,85 / 14,74 | 83.13 / 172.56 | 1.06 / 2.78 |
| LDT6 / HDT6 | 406,46 / 765,95 | 0,17 / 0,39 | 2,27 / 4,82 | 0,68 / 1,44 | 10.45 / 16.27 | 1.64 / 3.67 |
| USB0 | 1053,64 | 14,97 | 648,87 | 194,66 | 466.39 | 157.8 |
| USB1 | 899,39 | 9,2 | 355,07 | 106,52 | 218.14 | 64.99 |
| USB2 | 872,14 | 9,9 | 178,03 | 53,41 | 187.25 | 44.45 |
| USB3 / UAB3 | 914,04 / 1277,34 | 8,33 / 11,91 | 172,38 / 239,89 | 51,71 / 71,96 | 203.87 / 323 | 40.92 / 52.51 |
| USB5 / UAB5 | 821,83 / 1157,95 | 5,54 / 6,57 | 49,53 / 67,73 | 14,86 / 20,32 | 204.83 / 305.32 | 2.74 / 3.72 |
| USB6 / UAB6 | 847,25 / 1186,81 | 0,37 / 0,41 | 4,83 / 6,27 | 1,45 / 1,88 | 21.45 / 25.49 | 3.58 / 4.63 |
| MOT1 / MOT2 | 94,2 / 94,2 | 0,04 / 0,05 | 80 / 40 | 24 / 12 | 1298.98 / 1027.62 | 326.26 / 173.26 |
| MOT3 / MOT5 | 55,6 / 55,26 | 0,05 / 0,03 | 12 / 12 | 3,6 / 3,6 | 1027.62 / 533.51 | 173.26 / 97.75 |

In total, 70 vehicle categories are generated, which are doubled to 140 types of emission when considering urban and interurban travelling speeds. All these emission types apply to the different pollutants, which are shown in Table 6. In general, all emission factors decrease as the level of the EURO standard is increased. CO$_2$ emissions for gasoline vehicles are higher than diesel vehicles, this compound being the one with the least reductions since all technologies burn fossil fuels. Diesel vehicles contribute most of the emissions of PM$_{2.5}$, BC and NOx, but with important reductions when going from EURO 4 / IV to EURO 5 / V or EURO 6 / VI. It is interesting to note the high emission factors of PM$_{2.5}$ and CO from motorcycles, especially considering their recent increase in the Chilean vehicle fleet.

## 3.2 Annual emission trends at a national level

Using the activity levels and emission factors previously described, total emissions are calculated by pollutant, vehicle type, and country region. Table 6 shows a summary of the total annual emissions, with the variation percentages between 1990 and 2020. $CO_2$ has an increase of 207.7%, compared to 309% in mobility growth (VKT) for the same period. $CO_2$ official values for on-road transportation have been reported by Chile from 2010 until 2018 (MMA, 2018), differing by $\pm 1.4\%$ with values shown in Table 6 during those years.

Table 6. Total annual exhaust emissions produced by on-road transportation in Chile, 1990-2020

| | Year | | | | | | | (2020-1990)/1990 increase (+)/decrease (-) |
|---|---|---|---|---|---|---|---|---|
| | 1990 | 1995 | 2000 | 2005 | 2010 | 2015 | 2020 | |
| $CO_2$ [Tg yr$^{-1}$] | 8.6 | 12.4 | 14.6 | 14.9 | 19.2 | 22.1 | 26.6 | +207.7% |
| NOx [ton yr$^{-1}$] | 75.0 | 100.0 | 103.0 | 101.1 | 99.7 | 87.4 | 90.3 | +20.4% |
| BC [ton yr$^{-1}$] | 1.0 | 1.4 | 1.6 | 1.8 | 2.1 | 1.6 | 1.2 | +16.9% |
| $PM_{2.5}$ [ton yr$^{-1}$] | 2.0 | 2.7 | 2.8 | 3.0 | 3.1 | 2.3 | 1.7 | -12.5% |
| $CH_4$ [ton yr$^{-1}$] | 2.6 | 3.4 | 3.3 | 2.7 | 2.0 | 1.3 | 0.7 | -71.6% |
| CO [ton yr$^{-1}$] | 546.8 | 671.7 | 594.8 | 436.4 | 247.0 | 155.0 | 85.1 | -84.4% |
| VOC [ton yr$^{-1}$] | 54.6 | 67.9 | 59.6 | 43.2 | 25.1 | 15.0 | 6.8 | -87.5% |

Unlike $CO_2$, the rest of the local pollutants included in Table 6 are decoupled from the growth in mobility, reducing their contribution significantly thanks to technological improvements. NOx is the one with the least reduction compared with economic growth, with 20.4% of emissions in 2020 compared to 1990. Emissions of $PM_{2.5}$ and BC go up for the first 20 years of analysis, starting to decrease after 2010, where the reduction ratio of $PM_{2.5}$ is greater than BC. CO and VOCs, mainly associated with gasoline engines, show significant reductions, mainly due to the massive incorporation of three-way catalytic converters required for vehicles complying with EURO standards.

Table 7. Total annual exhaust emissions by vehicle type in Chile, 1990-2020

| | Category | Year | | | | | | |
|---|---|---|---|---|---|---|---|---|
| | | 1990 | 1995 | 2000 | 2005 | 2010 | 2015 | 2020 |
| $CO_2$ [Tg yr$^{-1}$] | LPV | 3.38 | 4.54 | 5.48 | 5.45 | 7.33 | 9.78 | 11.69 |
| | BUS | 1.23 | 1.62 | 1.84 | 2.12 | 2.59 | 2.37 | 3.09 |
| | TRUCK | 1.73 | 2.46 | 2.55 | 2.89 | 3.62 | 3.67 | 4.57 |
| | LCV | 1.62 | 2.69 | 3.52 | 3.55 | 4.82 | 5.40 | 6.12 |
| | TAXI | 0.67 | 1.13 | 1.20 | 0.90 | 0.84 | 0.85 | 1.07 |
| | MOT | 0.01 | 0.02 | 0.02 | 0.01 | 0.06 | 0.07 | 0.07 |
| $NO_x$ [ton yr$^{-1}$] | LPV | 21.44 | 25.22 | 24.92 | 21.77 | 16.51 | 13.14 | 10.81 |
| | BUS | 16.32 | 20.79 | 22.87 | 25.70 | 27.16 | 21.55 | 22.82 |
| | TRUCK | 21.07 | 29.39 | 30.05 | 33.06 | 37.15 | 32.91 | 33.05 |
| | LCV | 11.98 | 18.56 | 19.73 | 17.26 | 17.37 | 18.98 | 23.23 |

| | | | | | | | | |
|---|---|---|---|---|---|---|---|---|
| | TAXI | 4.14 | 6.05 | 5.45 | 3.36 | 1.49 | 0.75 | 0.33 |
| | MOT | 0.00 | 0.01 | 0.01 | 0.01 | 0.02 | 0.04 | 0.05 |
| $PM_{2.5}$ [ton yr$^{-1}$] | LPV | 0.13 | 0.17 | 0.19 | 0.27 | 0.37 | 0.31 | 0.25 |
| | BUS | 0.69 | 0.88 | 0.86 | 0.84 | 0.68 | 0.45 | 0.38 |
| | TRUCK | 0.99 | 1.37 | 1.36 | 1.39 | 1.23 | 0.82 | 0.58 |
| | LCV | 0.13 | 0.23 | 0.37 | 0.48 | 0.78 | 0.70 | 0.49 |
| | TAXI | 0.01 | 0.01 | 0.01 | 0.01 | 0.01 | 0.01 | 0.01 |
| | MOT | 0.02 | 0.03 | 0.02 | 0.02 | 0.04 | 0.03 | 0.02 |
| BC [ton yr$^{-1}$] | LPV | 0.08 | 0.10 | 0.12 | 0.18 | 0.25 | 0.20 | 0.13 |
| | BUS | 0.35 | 0.45 | 0.46 | 0.47 | 0.42 | 0.29 | 0.27 |
| | TRUCK | 0.52 | 0.74 | 0.75 | 0.80 | 0.78 | 0.54 | 0.39 |
| | LCV | 0.07 | 0.13 | 0.22 | 0.30 | 0.59 | 0.55 | 0.40 |
| | TAXI | 0.00 | 0.00 | 0.00 | 0.00 | 0.00 | 0.00 | 0.00 |
| | MOT | 0.01 | 0.01 | 0.01 | 0.01 | 0.01 | 0.01 | 0.01 |
| CO [ton yr$^{-1}$] | LPV | 360.98 | 400.45 | 346.54 | 254.48 | 135.47 | 83.53 | 43.98 |
| | BUS | 4.48 | 5.61 | 5.61 | 5.92 | 6.18 | 5.19 | 6.85 |
| | TRUCK | 6.11 | 8.41 | 8.51 | 8.77 | 9.13 | 8.05 | 9.77 |
| | LCV | 99.05 | 148.79 | 141.72 | 112.71 | 71.49 | 44.04 | 18.02 |
| | TAXI | 74.01 | 105.69 | 90.00 | 52.58 | 18.77 | 8.54 | 2.87 |
| | MOT | 2.15 | 2.77 | 2.44 | 1.94 | 5.96 | 5.66 | 3.62 |
| VOC [ton yr$^{-1}$] | LPV | 31.55 | 35.09 | 30.51 | 22.31 | 11.66 | 6.62 | 2.86 |
| | BUS | 1.50 | 1.88 | 1.80 | 1.77 | 1.49 | 0.97 | 0.73 |
| | TRUCK | 2.71 | 3.68 | 3.66 | 3.47 | 2.81 | 1.80 | 1.05 |
| | LCV | 11.12 | 16.47 | 14.45 | 10.13 | 5.60 | 3.26 | 1.08 |
| | TAXI | 6.47 | 9.28 | 7.93 | 4.66 | 1.65 | 0.71 | 0.18 |
| | MOT | 1.26 | 1.54 | 1.24 | 0.89 | 1.92 | 1.66 | 0.95 |
| $CH_4$ [ton yr$^{-1}$] | LPV | 1.36 | 1.56 | 1.47 | 1.17 | 0.78 | 0.49 | 0.28 |
| | BUS | 0.16 | 0.21 | 0.25 | 0.29 | 0.29 | 0.19 | 0.15 |
| | TRUCK | 0.21 | 0.30 | 0.35 | 0.41 | 0.47 | 0.33 | 0.21 |
| | LCV | 0.57 | 0.84 | 0.80 | 0.58 | 0.35 | 0.19 | 0.06 |
| | TAXI | 0.28 | 0.41 | 0.37 | 0.23 | 0.10 | 0.05 | 0.02 |
| | MOT | 0.02 | 0.02 | 0.02 | 0.02 | 0.04 | 0.03 | 0.02 |

Table 7 and Figure 5 show annual emission trends for six different compounds, divided by vehicle type. In the case of $CO_2$ and NOx, all types of vehicles have relevant contributions (Fig. 5a and 5b); diesel vehicles are responsible for the majority of $PM_{2.5}$ and BC emissions (Fig. 5c and 5d); and gasoline cars dominate CO and VOC emissions (Fig. 5e and 5f).

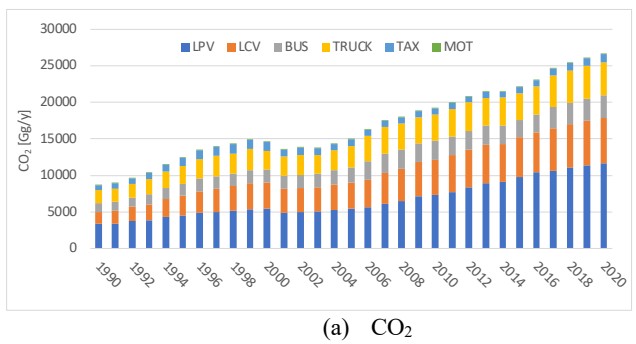

(a) $CO_2$

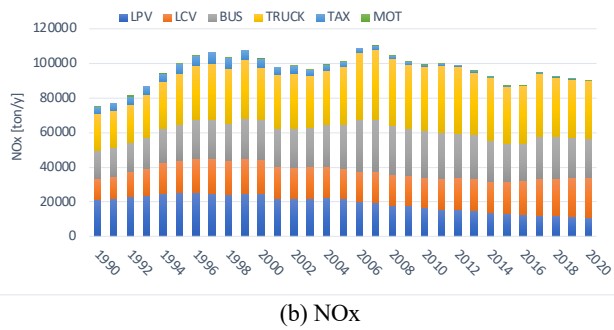

(b) NOx

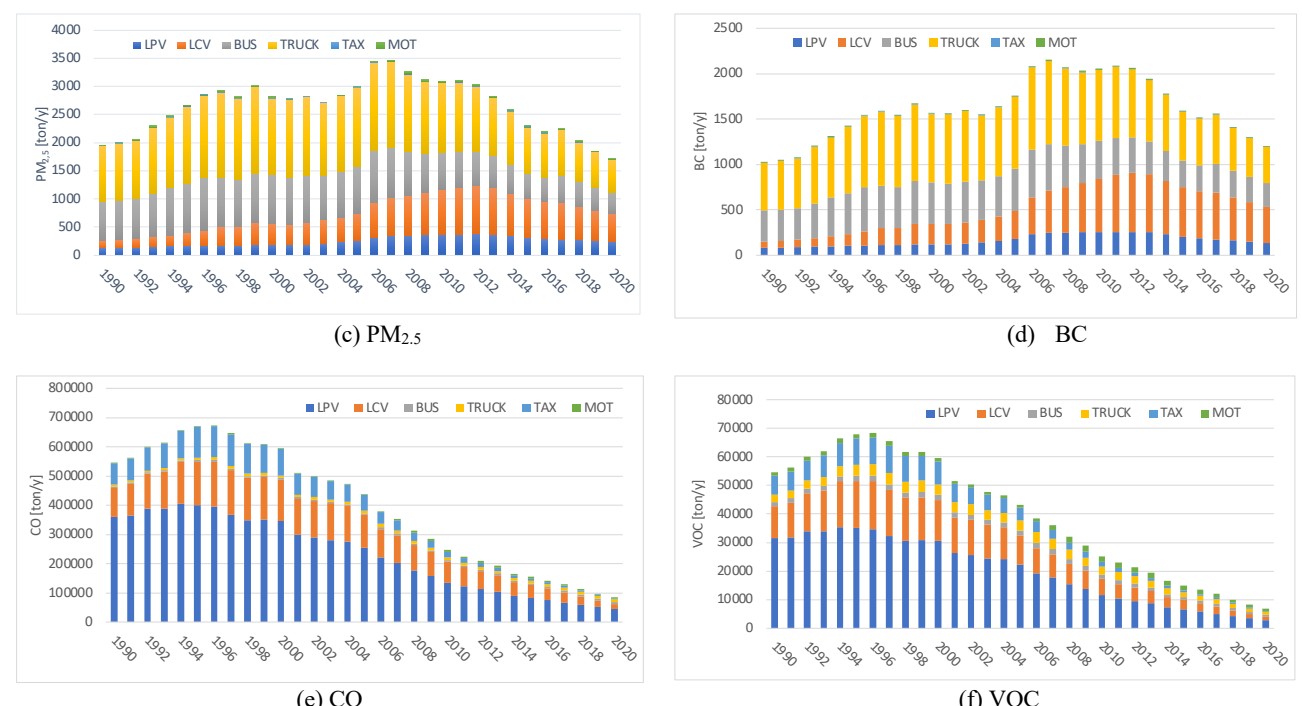

(c) PM$_{2.5}$

(d) BC

(e) CO

(f) VOC

Figure 5. Total annual exhaust emissions trends according to vehicle type in Chile, 1990-2020

Except for CO and CO$_2$, all annual emission curves show continuous growth from 1990 to 2006-2008. Thereafter, the general trend for local pollutants is to decrease. This turning point coincides with the change throughout the country in the requirement from EURO 1 to EURO 3 for light vehicles and from EURO II to EURO III for heavy-duty vehicles between 2005 and 2006 (Table 2). The limits for light gasoline vehicles for NOx, CO and VOC between EURO 1 and EURO 3 are strict, which largely explains this change in trend. The change in PM$_{2.5}$ in 2007 occurs immediately after the year 2006, when EURO III was imposed for buses in the most important regions of the country and EURO II was required for the first time from the rest of the public and private buses (previously many of them had no obligation to meet standards). Subsequently, during the years 2011-2014, EURO 4 and EURO 5 were required for light vehicles and EURO IV and EURO V for buses and trucks, which has allowed to continue reducing emissions despite the growth of the vehicle fleet.

It is important to note that Chile has one of the most advanced state-own vehicle homologation centres in Latin America (3CV Centro de Control y Certificación Vehicular, https://www.mtt.gob.cl/3cv.html), which controls the entry of all new vehicles sold in the country since 1996. 3CV has an emissions laboratory that allows experimental verification of compliance with the regulations in force in Chile, according to European or United States procedures. Additionally, since 2007 the periodic technical inspection (PRT) procedure in Chile includes tests under load and simultaneously controls CO, HC and NOx for light vehicles, and opacity for buses and trucks. The combination of both procedures, 3CV and PRT, are the main policy tools

to enforce that the emissions of the vehicles that circulate in the country comply with the regulations in force at the time of their sale and during their life span.

From 2008 onwards, BC emissions are not mitigated with the same rate as PM$_{2.5}$ does, the BC reduction being less effective, which can contribute to local health problems and negative effects on local climate change (Bond et al., 2013; WHO Regional Office for and Europe, 2012). This can be explained given that the BC / PM$_{2.5}$ fraction for light vehicles decreases significantly when going from EURO 4 to EURO 5, but this is not the case for heavy vehicles, which have this significant decrease later, between EURO V and EURO VI (Table 4). Trucks and buses are the main contributors to the total emissions of BC, and these 45 two categories are the latest to be required with EURO VI. According to the information given in Table 4, the BC fraction in PM$_{2.5}$ emissions for heavy duty trucks is 75% in EURO V, which is not a considerable reduction compared to other vehicle types.

Finally, all the emission curves show a drop in the 1999-2004 period, which is explained by the impact that the Asian financial 50 crisis had on Chile, significantly affecting the sale of motor vehicles and their activity.

### 3.3 Spatial disaggregation

The complete database for the period 1990-2020 is available at doi: http://dx.doi.org/10.17632/z69m8xm843.2. This inventory is part of the first gridded national inventory of anthropogenic emission for Chile of criteria pollutants as well as GHG (hereafter INEMA from Spanish Inventario Nacional de EMisiones Antropogénicas), presented by Alamos et al (2022). 55 INEMA comprises emissions for vehicular, industrial, energy, mining and residential sectors for the period 2015-2017 in Chile.

The spatial disaggregation of emissions at the national level shows the high concentration of emissions in urban areas and main roads in the country. Figure 6 shows the fraction of PM$_{2.5}$ emissions for the year 2020 over the cells of 0.01x0.01 degrees of latitude and longitude, which is equivalent to approximately 1.11x1.11 kilometres. It is difficult to clearly identify these 60 emissions on the complete map of Chile, due to its shape (Fig. 6a). However, when zooming in on each region, the populated areas with high emissions on the road network appear. Fig. 6b shows the city of Antofagasta, which is approximately 22 kilometres long and has a population of 388 thousand inhabitants, which concentrates most of the on-road vehicle activity. Fig. 6c corresponds to the Metropolitan Region with Santiago in the centre, where 7 of the almost 19 million inhabitants of Chile live (INE, 2020). The same image shows the city of Valparaíso on the coast and Rancagua, south of Santiago. Finally, 65 Fig. 6d shows the Bio-Bio region, with the city of Concepción that is home to 221 thousand inhabitants.

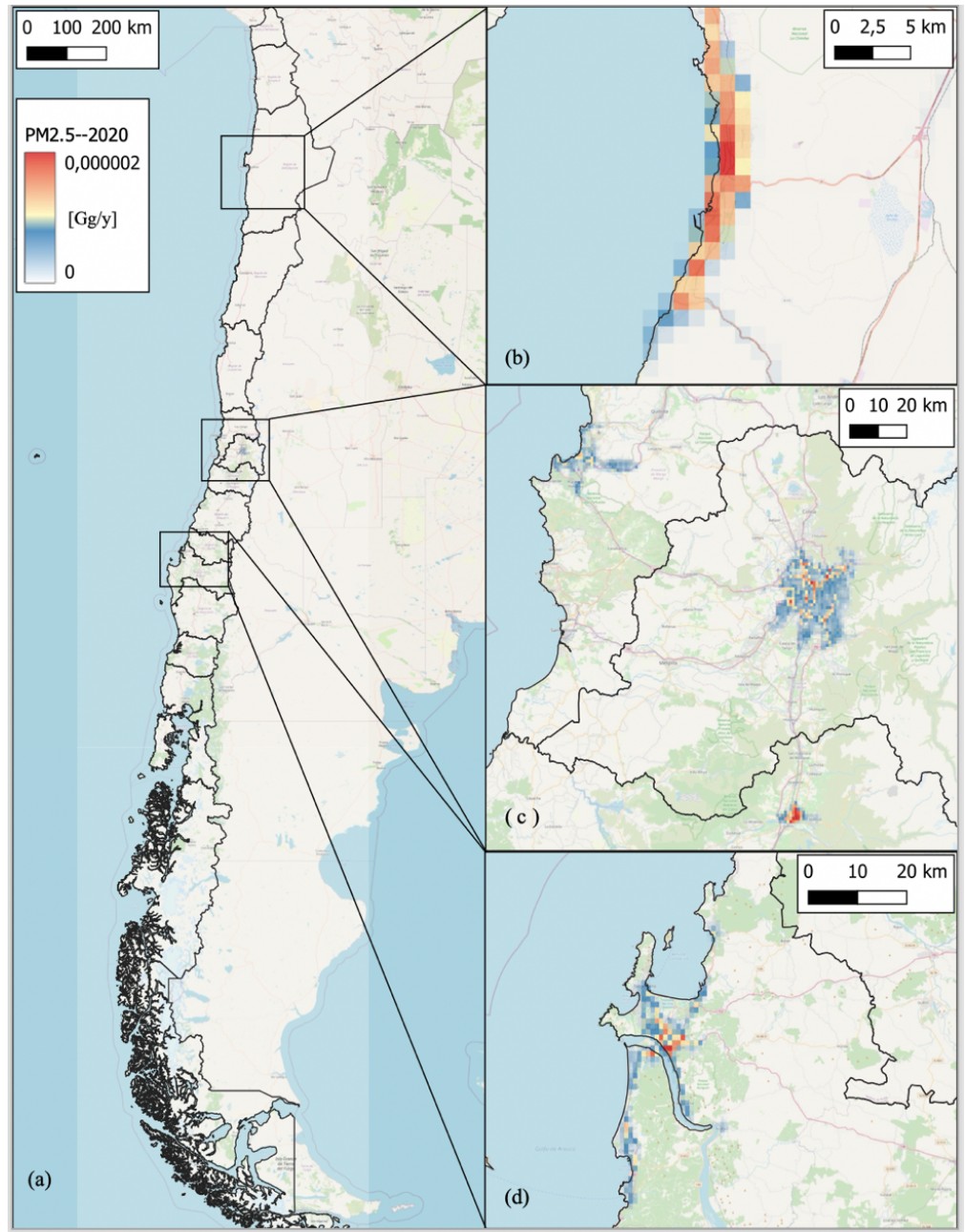

Figure 6. Spatial disaggregation of PM$_{2.5}$ exhaust emissions along the country (a), Antofagasta (b), Metropolitan Region (c) and Bío-Bío/Concepción (d), 2020, as a fraction of Gg/y.

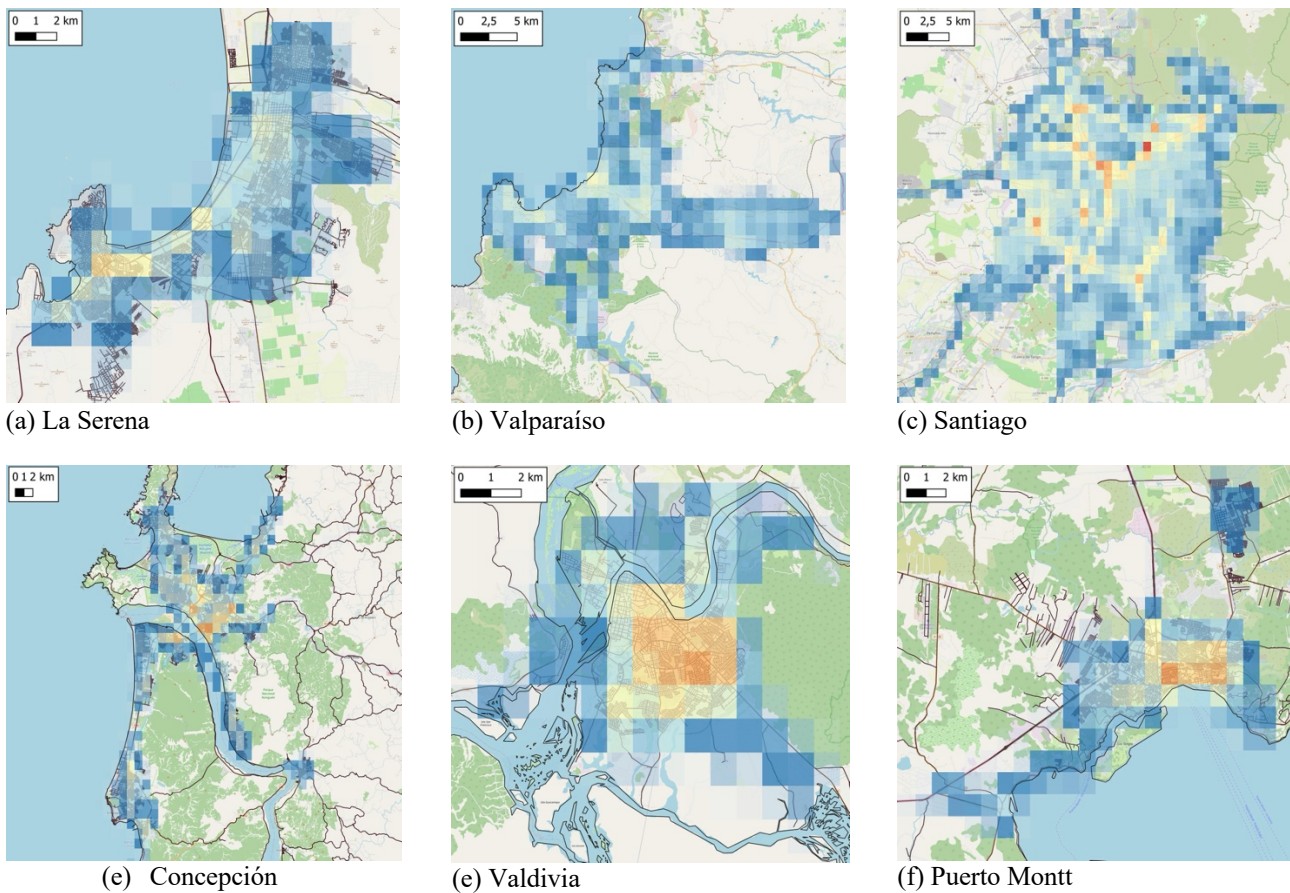

(a) La Serena    (b) Valparaíso    (c) Santiago

(e)  Concepción    (e) Valdivia    (f) Puerto Montt

Figure 7. Spatial disaggregation of NOx exhaust emissions for six cities in Chile, 2020, as a fraction of Gg/y. Colour legend same as Figure 6.

The images in Figure 7 show the fraction of NOx emissions for the year 2020 in six major cities of Chile, from north to south. The emission grid is superimposed with the road network, green areas, and uninhabited areas, obtaining a good matching between them. In each city, the areas of greatest activity are coloured with warmer tones, indicating greater NOx emissions, decreasing towards colder tones for a lower fraction of these emissions.

### 3.2.4 Comparison with previous results

A direct comparison of emissions from this study with other emissions estimates was performed to reflect the differences in estimation approaches between local (bottom-up) and global (top-down) models, as well as the sensitivity to different assumptions in the estimates. Figures 8, 9 and 10 show the comparison among local estimates from this work - INEMA, the National Emissions Inventory – INGEI (MMA, 2020) and an estimate using the LEAP model (Kuylenstierna et al., 2020); and

global estimates by the EDGAR V5.0 global model (Janssens-Maenhout et al., 2017) – EDGAR, the CAMS-GLOB-ANT v. 4.2 dataset – CAMS (Granier et al., 2019), and the CEDS dataset – CEDS (McDuffie et al., 2020; Smith et al., 2015), for $CO_2$,

$CH_4$, PM, BC, CO, and $NO_x$ from 1990 to 2020, according to the pollutants available in each estimate. It is worth mentioning that EDGAR, CAMS and CEDS are not independent. For historic years CAMS is mostly based on EDGAR but extrapolated to more recent years, using other information such as trends from CEDS.

$CO_2$ and methane emissions are compared in Figure 8. There is a good agreement in $CO_2$ emissions and trends among most of

the estimates for most of the period, which indicates that the activity level, i.e. fuel consumption, is consistent between top-down and bottom-up approaches. The largest difference is observed for the LEAP inventory from 2015, caused by a sudden reduction in 2015 and a slower increase between 2015 and 2020. Methane emissions show similar trends but different levels between this work (INEMA) and EDGAR from 1990 to 2004, global estimates being higher than the local estimate by 20% to 43%. The trends became divergent since 2005, with decreasing emissions in the local estimate and increasing emissions in

EDGAR, CAMS and CEDS. EDGAR and CAMS estimates were very similar between 2000 and 2011. Later, CAMS estimates increased linearly and more slowly than EDGAR emissions. On the other hand, EDGAR and CEDS estimates were the same between 2000 and 2014. Later, CEDS increased slightly more slowly than EDGAR. The decrease in the CEDS estimate between 2019 and 2020 is not reported in EDGAR.

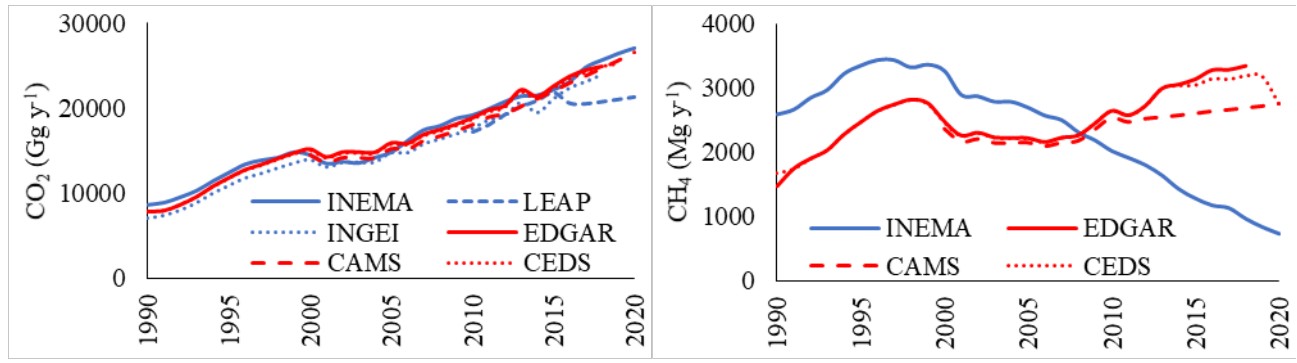


Figure 8. Comparison between $CO_2$ (left) and $CH_4$ (right) between this work (INEMA) and other local and global emission inventories.

Figure 9 shows emissions estimates for PM and BC. With the exception of CEDS, there was a good agreement between local

and global emissions inventories between 1990 and 1998. However, EDGAR and CAMS show a sudden increase in 1999 that cannot be explained by a change in activity and is likely due to a change in the emission factors used in those inventories. Furthermore, after 1999, these global inventories show a consistent increasing trend. Such trend that was not followed by local estimates, which show a stabilization between 1997 and 2005, and a rather consistent decrease since 2007. As a result, EDGAR and CAMS PM (BC) emissions from 1999 to 2015 were between 85% (87%) and 315% (208%) higher than those from the

local inventory. On the other hand, CEDS estimates for BC were even higher than EDGAR and CAMS estimates for the whole period, although they followed similar trends between 2000 and 2015. CEDS/INEMA BC emission ratios range from 2.76 to 5.69, suggesting that BC emission factors in the CEDS dataset are significantly higher than those used in this work. Since this work's emission factors are based on the COPERT model and the actual vehicle technology distribution, higher PM and BC emission factors used in EDGAR and CEDS imply assumptions of an older fleet in global inventories. Standards for diesel

vehicle emissions and sulphur fuel content have been greatly improved in Chile since 2000, so EDGAR, CAMS and CEDS emissions and increasing trends for PM and BC are likely overestimated.

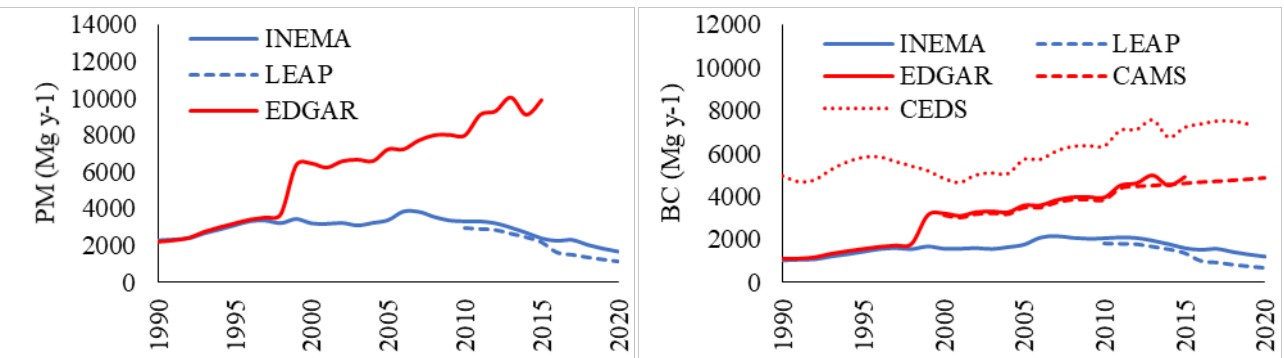

Figure 9. Comparison of PM$_{2.5}$ (left) and BC (right) emissions between this work (INEMA) and other local and global

inventories.

CO and NOx emissions and trends are shown in Figure 10. CO emissions were considerably higher in the global inventories (EDGAR, CAMS and CEDS) than in the local inventories (INEMA and LEAP), with a mean difference of 254% [95% - 811%], and the trends were divergent since 2006, with increasing emissions in the global inventories and decreasing emissions

in the local inventories. This is likely due to assumptions of an older fleet and, therefore, higher CO emission factors in the global inventories. A similar situation was observed for EDGAR and CEDS NOx emissions compared to local estimates, which showed rather similar levels between 1990 and 2005, with larger differences between 1993 and 1998, and diverged after 2005, with increasing emissions in the global inventories and decreasing emissions in the local inventories. Differences increased from 9% in 2009 up to 70% in 2015, with respect to local inventories. Once again, these differences suggest that

global estimates did not reflect improvements in Chile's vehicle fleet after 2005.

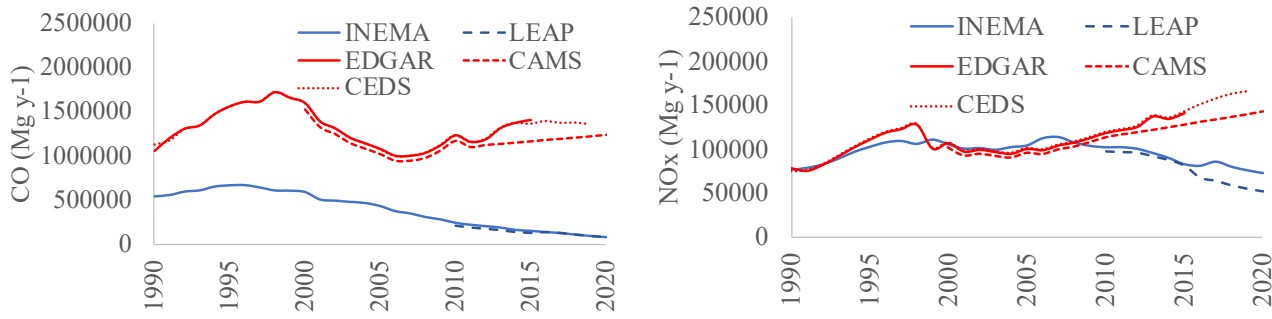

Figure 10. Comparison between CO (left) and NOx (right) between this work (INEMA) and other local and global emission inventories.


Finally, the CO/NOx ratio and its trends are shown in Figure 11, which not only includes Chile but also a comparison with European and two other countries in the Latin America and the Caribbean (LAC) region between 1970 and 2020. The CO/NOx ratio was much higher in the global inventories than in the local inventories, with a mean difference of 209% [90% - 457%] between EDGAR and INEMA estimates for Chile, and shows a decreasing trend in both, with more fluctuations in the global

inventory. The differences in emissions and trends for CO and NOx suggest that global emission inventories use emission factors that correspond to technologies older than those that have been and are currently used in Chile. Considering the differences between EDGAR data and this study's results for Chile, trends in CO/NOx ratio for other European and LAC countries from EDGAR were included. A big difference appears between these two groups of fleets, the CO/NOx ratio being much higher for LAC selected countries. In other words, according to EDGAR figures, LAC CO/NO ratios reach European

values 40 years later (1970 versus 2012), which seems inaccurate according to local estimates. Chile's CO/NOx ratios are in the same range as those found in European countries, which is supported by the fleet renewal shown in Figures 3 and 4. Most of the Chilean fleet consists of Euro II/2 and Euro III/3 vehicles, which have much lower CO/NOx ratios that pre-Euro ones. Our analysis for Chile suggests that a careful analysis of national versus global estimates and/or emission factors for road transport emissions is needed for other LAC countries as well.


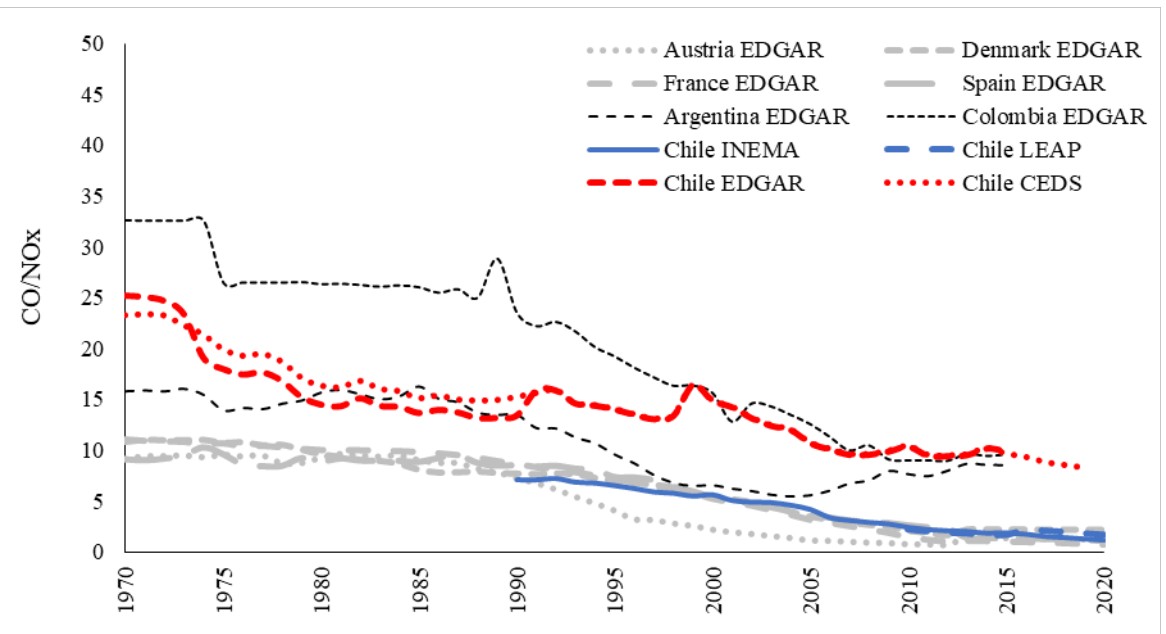

Figure 11. CO/NOx ratios for Chile and different countries from EDGAR and this work.

External data obtained from models EDGAR/CEDS/LEAP

**Conclusions**

This paper describes an original dataset for transport emission in Chile between 1990 and 2020, spatially distributed at 0.01° x 0.01°. The dataset is based on annual reports from governmental agencies, and estimates the evolution of air pollutants (CO, VOC, NOx, $PM_{2.5}$), greenhouse gases ($CO_2$, $CH_4$) and black carbon (BC). Results were contrasted with EDGAR, CAMS and CEDS datasets.

The analysis shows a significant growth of the vehicle fleet coupled with increasing $CO_2$ emissions, which agree with the national inventory of GHG. Air pollutants show different patterns, with a general decreasing trend which coincides with pollution control measures. Data shows a clear relationship between these emissions and the introduction of better fuel quality, due to reduction of sulphur content, and enforcement of technological improvements. These policy measures included regulation of emission standards for new vehicles into the fleet, mandatory periodic technical inspection for in-use vehicles, as well as effective procedures for regulation enforcement.

The comparison with EDGAR, CAMS, CEDS, and locally estimated datasets shows agreement in $CO_2$ estimations and striking differences for local compounds, with global estimates consistently higher. This disagreement is likely due to differences in assumptions of vehicle technologies characterizing the fleet and quality of the fuel used. In the case of PM and BC trends between EDGAR and this transport dataset diverge from 1998, for CO, NOx and $CH_4$ since 2006 -2008. Results suggest that

global emission inventories use emissions factors that do not coincide with the technologies of the vehicle fleet. EDGAR assumes a 40-year delay in technological update for Latin-American vehicle fleets compared to European ones, which is inaccurate for the case of Chile according to the dataset presented in this paper.

Every dataset has limitations and this is not an exception, INEMA does not include cold start emissions, neither consider a calibration of fuel consumption according to vehicle age. The use of international emission factors is a second best compared to using locally measured emission factors and COPERT does not cover aging for all vehicle categories in the dataset. The impact of COVID-19 is not considered in 2020, but other studies have addressed these effects on urban emissions in Santiago (2020). However, these limitations should not significantly change the results of the paper since the database provided is more accurate and extended than the existing ones, and the comparative analysis with external datasets show differences that need attention.

This paper illustrates the potential of local datasets for policy ex-post impact assessment. It also reinforced the value of available official raw data, produced with transparent methods and on a regular basis, as well as the production of national inventories. Further work could build on the dataset presented in this paper to produce projections and scenarios for future policy making. Work should be done on the construction of local emission factors, this is the only information of the modelling that is not produced locally, real emissions campaigns of a sample of the fleet could strengthen the results of this analysis.

## Data availability

This dataset contains annual exhaust emission inventories of CO, VOC, NOx, PM2.5, CO2, CH4 and BC from on-road transportation in Chile, for the period 1990–2020. The data is presented as NetCDF4 files, in Gg/y per cell for each specie and year, gridded with a spatial resolution of 0.01° x 0.01° covering the domain 66°–75° W and 17°–56° S. It can be accessed through the open access data repository http://dx.doi.org/10.17632/z69m8xm843.2, under a CC-BY 4 license (Osses et al., 2021).

## Author contributions

NR: Methodology, Investigation, Writing - Review & Editing; CI: Methodology, Conceptualization, Writing - Review & Editing; VV: Conceptualization, Formal analysis; IL: Methodology, Formal analysis, Data Curation, Visualization; NP: Data Curation, Visualization; DO: Data Curation, Visualization; KB: Methodology, Review & Editing; ST: Methodology, Review & Editing; NH: Review & Editing; LG: Review & Editing; BG: Methodology, Data Curation; MO: Conceptualization, Methodology, Investigation, Writing - Review & Editing.

## Competing interests

Author MO is a guest member of the editorial board of the journal.

**Acknowledgments**

The authors would like to acknowledge to the data providers of EDGARv5.0 database, available at their air pollutant website (https://edgar.jrc.ec.europa.eu/overview.php?v=50_AP). Also, we acknowledge developers of the CAMS dataset, described by Granier et al. (2019). Finally, data provided by the GEIA data portal ECCAD (https://eccad.aeris-data.fr/) has been a great support for this study.

**Financial support**

Institutional support and funding have been kindly provided by Center for Climate and Resilience Research FONDAP #15110009, Technological Scientific Center of Valparaíso ANID PIA/APOYO AFB180002 and the EU project Prediction of Air Pollution in Latin America and the Caribbean PAPILA, ID: 777544, H2020-EU.1.3.3.

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
