# Peer review of "High-resolution spatial distribution maps of road transport exhaust emissions in Chile, 1990 - 2020."

_Earth System Science Data, 2021_

## Author Comment (AC1)

**Author's response**

Dear Editor,

Thank you for reviewing our manuscript number ESS-2021-2018. We have gone through all the referee's commentaries and adjusted the manuscript accordingly. After this paragraph you will find our responses. The style used in the response letter is the following: the original general comments made by the referee are kept in normal text (initiated with R), our responses are *in blue italics initiating with A* (Authors). The corresponding edit in the manuscript will be included in red.

**Referee comments 1 (RC1)**

Anonymous Referee #1, 09 Sep 2021

High definition spatial distribution maps of on road transport exhaust emissions in Chile 1990-2020; Osses et al.; essd-2021-218.

R1: This manuscript describes the methodological aspects in preparing a high resolution (0.01°x0.01°) inventory of road transport emission for Chile for years 1990-2020. It includes GHG gases ($CO_2$ CH4) and air quality pollutants (CO, VOC, NOx, PM, and BC). Special emphasis is given to latter one. It considers the impact of changing emissions standards in emissions trends. The analysis includes a comparison with international EDGAR data set, showing good agreement in CO2 but important differences in SLCP.

R1: General comments

R1: The comparison with EDGAR is a very important and useful analysis that benefit the international inventory community to achieve better and reliable global emissions models. A good/plausible explanation is given for the encountered differences with EDGAR.

*A: Given the importance of the comparison highlighted by the Reviewer, the analysis has been expanded (see next answer) and additional explanations have been added to the text in section 3.2.4.*

R1: In this line it is recommendable that the authors also include a comparison with the **Community Emissions Data System (**CEDS). Moreover, to better emphasizes the uncertainties level of the proposed inventory, it should show, if possible, in a summary table, other emissions calculated for Chile, either national/regional or by cities, for GHG and/or SCLP if available.

*A: We appreciate the comment since we fully agree it is important to reinforce the comparison analysis. Thus, in addition to EDGAR, CEDS and CAMS datasets for Chile have been included in section 3.2.4, with extended comments explaining similarities and differences. Additionally, the official inventory reported by the Chilean government for GHG (INGEI) and a national estimate of transport emissions using LEAP model have been added to the comparison analysis. In summary, our emission inventory for exhaust on-road transportation emissions (INEMA) is compared with two local national inventories (INGEI, LEAP) and three global models (EDGAR, CAMS, CEDS).*

R1: The manuscript is well written, is suitable for the inventory special issue and is acceptable for publication after some minor revisions and additional comments.

*A: We appreciate the recommendation and performed a complete general revision to the whole final text, adding minor revisions and to integrate the new comparisons and comments.*

R1: Other comments:

R1: Line 60 page 8

R1: Determining the active fleet is always a complicated matter, especially when a long time series is calculated. The calculation of active fleet should include deregistration and scrapped rate for each vehicle cohort. This produces that a new registered vehicle in year n will be out of the roads in year n + m (number of active years). Since you are using (new?) registered vehicles, have you estimated how many years each (type of) vehicles with technology j are active? Also fuel consumption an emission factors degrade with aging vehicles. Have you considered any emission factor and fuel consumption function correction for each cohort? Also, VKT may be affected by aging vehicles. Although you calibrate the number of vehicles by fuel sales, some comments should be said with respect to the above point. How are the numbers of vehicles estimated in Figures 2?

*A: We did not use new registered vehicles. Data provided by INE corresponds to annual registration of vehicles, i.e., the vehicles that each year pay their circulation permit after having approved the periodic technical inspection (explained in line 326, page 17). We added this explanation to the methodological section, in sub-section 2.1.*

*Using annual registration of in-use vehicles partially solves the problem of deregistration and scrapped rate for each vehicle cohort. However, some of the vehicles with annual circulation permit may be not used, may have very low circulation rates, or may be used in a different region of that of their registration. To consider these issues, we contrasted calculated fuel consumption (TFC) with real fuel sales by region (line 158), and used a correction factor to adjust the number of registered vehicles in each region, as explained in sub-section 2.2 of the methodology.*

*Regarding the emission factors degrading with age, we used COPERT emission factors which consider aging factors for some of the categories and emission components. We added a line explaining this in the subsection 2.3 of the methodology: "COPERT V considers correction of emission factors by vehicle age for light vehicle categories EURO 3 & 4 and for VOC, CO, NOx. These corrections were also applied".*

*We did not calibrate fuel consumption by vehicle age. We acknowledge these limitations of the dataset in the conclusions in a new paragraph.*

R1: Some additional considerations should be added with respect to changes in mobility indicators since these are mentioned in the results. Number of vehicles per household, number of vehicles per inhabitants, number of vehicles/GDP per capita and so on. This extra information, although not strictly necessary will enrich your paper and analysis.

*A: Thanks for the comment. We considered mentioning some of these general mobility indicators in the description but decided not to do it. The paper covers 30 years and most of these parameters change each year and by region, making difficult to provide a clear summary of information. We agree this information is not strictly necessary, it would enrich the analysis, but it might confuse the readers since those mobility indicators are not part of the inputs for our emission model.*

R1: Line 110 page 9:

R1: The English should be rephrased, probably the word "between" is not correct and may be replaced by "among". You are distributing the region's emissions proportionally to the population density of that region. What is the finest population density scale available in your calculation? Are the roads weighting factors constant to all regions in Chile?

*A: Yes, now on page 10 line 218 it should read "among". The finest population density corresponds to a city according to definition of the National Statistics Institute (INE), i.e., an urban area with more than 5000 inhabitants. The weighting factors vary by region, urban and interurban areas, and by city, and they were provided by the Transport Secretariat, SECTRA (Osses et al, 2010). This explanation and its reference have been added at sub-section 2.4 of the methodology, lines 214-215, page 10.*

R1: Although the spatial disaggregation's methodology is in general understandable, some extra details should be added. It needs some extra clarifications, with regards to the spatial scales. Emissions are calculated in "Regions", then is downscaled to what? … Districts? -> Municipalities? … How do you derive urban from non-urban areas? Are the roads weights similar in urban / rural areas? Readers may profit from the methodology used in your calculations.

*A: Regions are downscaled to urban and interurban roads using the data provided by SECTRA, and then urban areas are downscaled to cities in the region, with are urban areas with at least 5000 inhabitants. This explanation has been added to sub-section 2.4 of the methodology, lines 220-223, page 10.*

R1: Line 145 page 12

R1: Check typo error "if" : The vehicles in category if heavy diesel…"

*A: Yes, it should say "of", it was mended.*

R1: Figure 6: Caption should declare the emissions color scale (e.g. "same as Figure 5") o added to the figure.

*A: Thanks for the observation, "same as Figure 5" was added to the caption.*

R1: Figure 10, page 22. The Figure shows CO/NOX ratios for other countries. Please define the references for these data.

*A: The references correspond to the three global databases used for comparisons (EDGAR/CAMS/CEDS). This has been added to the figure caption.*

---

## Author Comment (AC2)

**Author's response**

Dear Editor,

Thank you for reviewing our manuscript number ESS-2021-2018. We have gone through all the referee's commentaries and adjusted the manuscript accordingly. After this paragraph you will find our responses. The style used in the response letter is the following: the original general comments made by the referee are kept in normal text (initiated with R), our responses are *in blue italics initiating with A* (Authors). The corresponding edit in the manuscript will be included in red.

**Referee comments 2 (RC2)**
Anonymous Referee #2, 13 Sept 2021

R2: This manuscript presents a description of a Chilean high resolution gridded emission inventory of road transport exhaust emissions for the period 1990–2020, as well as a comparison against the emissions reported by the EDGAR inventory. As stated by the authors in the introduction section, the availability of high-resolution emission inventories in Chile that are consistent, updated and cover a long period of time is currently limited. Therefore, the dataset presented in the manuscript if of interest and a good contribution to ESSD. I recommend the manuscript to be published once the following comments have been addressed:

R2: Title of the manuscript: I would suggest to rephrase the title from "High-definition spatial distribution maps of (···)" to "High-resolution spatial distribution of (···)" as it is more frequently used in the scientific literature.

*A: The title has been modified according to the suggestion.*

R2: Vehicle fleet composition: According to the authors, information on the vehicle fleet composition per political region is obtained from official government data. Is this source of information reporting data on registered vehicles or the actual "in-use fleet" (i.e., on-the-road or circulating fleet)? Several studies have highlighted strong discrepancies between registered and in-use vehicle fleet compositions. Official vehicle registries can suffer from certain limitations, including: i) they may include vehicles that have been scrapped (or that are registered but hardly being used) and ii) they include information regarding where the vehicles are registered but not where are actually driven. How did the authors overcome these limitations? Please provide an explanation.

*A: Data provided by INE corresponds to annual registration of vehicles, i.e., the vehicles that each year pay their circulation permit after having approved the annual technical revision. We added this explanation to the methodological section, in sub-section 2.1.*

*Using annual registration of in-use vehicles partially solves the problem of deregistration and scrapped rate for each vehicle cohort. However, some of the vehicles with annual circulation permit may be not used, may have very low circulation rates or may be used in a different region of that of their register. To consider these issues, we contrasted calculated fuel consumption (TFC) with real fuel sales by region (line 155), and used a correction factor to adjust the number of registered vehicles in each region, as explained in sub-section 2.2 of the methodology.*

R2: Total Fuel Consumption (TFC): Could you provide a figure (or summary table) that shows the results of the comparison between calculated TFC and reported fuel sales for each region? This would allow understanding better the discrepancies between the two datasets.

*A: Figure 2 has been added to the main text, showing the difference between official fuel sales and estimated total fuel consumption, for gasoline and diesel.*

R2: Spatial distribution: Could you provide a reference for the toll barrier vehicle counts used for computing the average road weight factors? Could you provide a summary table with the shares regarding the distribution of vehicles into urban and interurban activity per region? Perhaps this information could be included as part of Table 3 (Annual activity level not only per region and vehicle type but also discriminated between urban and interurban).

*A: The reference (MOP, 2020) was added (line 212, page 10) and the official link has been included at the References section.*

R2: Emission factors: Authors use the emission factors reported by COPERT 5, which is a vehicle emission calculator originally developed for Europe. Can the authors say something on how precise is COPERT in reflecting the Chilean fleet and driving conditions? Is there any database of measured local emission factors that could be used for comparison purposes?

*A: Unfortunately, Chile does not have a robust database of local emission factors covering all existing vehicle technologies and driving conditions. There are some local measurements using dynamometer facilities as well as portable emission measurement systems, but not enough for supporting a national emission model, particularly for newer technologies such as EURO 5/6. The Chilean homologation process allows both US and Europe-based emission standards, but most of the vehicles are certified with EURO standards. For this reason, COPERT has been accepted as an appropriate international model by Chilean researchers and authorities (Osses, 2010; MMA, 2014; Osses, 2014; Tolvett, 2016; Gallardo, 2018; Mazzeo, 2018; Huneeus, 2020).*

R2: Cold-start emissions: Are cold-start emissions included in the inventory? These type of exhaust emissions could be significant in certain regions of the country during winter time. Please specify.

*A: Cold-start emissions effect was not considered and this was added to the conclusions regarding limitations of the dataset, as proposed by the Reviewer.*

R2: Comparisons with EDGAR (1): At the beginning of section 3.2.4, authors mention that they performed a comparison between INEMA and EDGARv4.3.2. However, it looks to me that the comparison is done against EDGARv5.0, as v4.3.2 reports emissions only until 2012, and v5.0 up to 2015. Please specify and correct if needed.

*A: The observation is correct; the dataset corresponds to EDGARv5.0 and it has been corrected through the text and references. The following references where updated:*

*Crippa, M., Guizzardi, D., Schaaf, E., Solazzo, E., Muntean, M., Monforti-Ferrario, F., Olivier, J.G.J., Vignati, E.: Fossil CO2 and GHG emissions of all world countries - 2021 Report, in prep.*

*Crippa, M., Solazzo, E., Huang, G., Guizzardi, D., Koffi, E., Muntean, M., Schieberle, C., Friedrich, R. and Janssens-Maenhout, G.: High resolution temporal profiles in the Emissions Database for Global Atmospheric Research. Sci Data 7, 121 (2020). doi:10.1038/s41597-020-0462-2.*

R2: Comparisons with EDGAR (2): The discrepancies between the emission trends reported by INEMA and EDGAR are quite significant, especially for NOx. In my opinion, it would be good to include in the comparison other state-of-the-art global emission inventories such as CEDS (http://www.globalchange.umd.edu/ceds/) or ECLIPSEv6b (https://iiasa.ac.at/web/home/research/researchPrograms/air/ECLIPSEv6b.html), in order to see if their trends match better with the one reported by INEMA. Moreover, both CEDS and ECLIPSE report emissions up to more recent years (e.g., 2019).

*A: We appreciate the comment since we fully agree it is important to reinforce the comparison analysis. Thus, in addition to EDGAR, CEDS and CAMS datasets for Chile have been included in section 3.2.4, with extended comments explaining similarities and differences. Additionally, the official inventory reported by the Chilean government for GHG (INGEI) and a national estimate of transport emissions using LEAP model have been added to the comparison analysis. In summary, our emission inventory for exhaust on-road transportation emissions (INEMA) is compared with two local national inventories (INGEI, LEAP) and three global models (EDGAR, CAMS, CEDS).*

R2: Comparisons with EDGAR (3): Regarding the discrepancy between the NOx emission trends reported by INEMA and EDGAR, and considering that road transport is the main contributor to total NOx emissions, perhaps it would be interesting to contrast these results against the evolution of NO2 concentrations in traffic stations for the same period of time (i.e., 1990 to 2015). These would allow seeing if NO2 concentrations show a positive or negative trend (or if concentrations remain unchanged) and subsequently if they correlate better with the trend reported by INEMA or EDGAR.

*A: Thank you for the suggestion. Unfortunately, the air quality monitoring network along continental Chile (https://sinca.mma.gob.cl/) does not provide a national coverage of nitrogen dioxide data. Stations outside Santiago do not provide NO2 except for a few sites, and the period covered in those sites is too short to establish long-term trends. Except for mass concentrations of particles, and to some extent sulfur dioxide, the coverage is poor for other pollutants. Another issue is identifying traffic dominated stations. Stations are placed to monitor the compliance of air quality standards set for protecting human health, and not for process understanding. One could try to minimize the effect of residential sources by considering summer values, and rush hours to capture traffic emissions (See Gallardo et al, 2012). This would be feasible but for a few places, and that would not be particularly helpful identifying national emission trends. In previous work, Menares et al (2020) analyzed NO2 trends from in situ data in Santiago and found increasing trends for the period 2001-2018 over Eastern Santiago, which the authors attribute to changing photochemical regimes.*

*In a recent work, Goldberg et al (2021) estimated urban NOx emissions trends for the period 2005-2019 using satellite borne measurements of the NO2 column. Over Santiago they infer increasing emission trends between 2005 and ca. 2012 and declining trends thereafter (See image extracted from the paper). Previously, Duncan et al (2016) estimated a very high trend (30±17%) in the NO2 column as observed from the Ozone Monitoring Instrument (OMI), in some agreement with in-situ data (Menares et al, 2020). The same data but considering the period between 2005 and 2020 results in a small and insignificant trend ( -3.13±12.4%*), possibly due to considering the pandemic and the political unrest after October 2019.*

*Thus, all in all, at this point it appears difficult to resolve the inconsistencies in trends inferred from different data and methodologies. Regional scale modeling studies will provide further insights in the matter, but that of course, is beyond the scope of this paper.*

[Figure]

*Duncan, B. N., Lamsal, L. N., Thompson, A. M., Yoshida, Y., Lu, Z. and co-authors. 2016. A space-based, high-resolution view of notable changes in urban NOx pollution around the world (2005–2014. J. Geophys. Res. Atmos. 121, 976–996. doi: 10.1002/2015JD024121*

*Gallardo, L., Escribano, J., Dawidowski, L., Rojas, N., de Fátima Andrade, M., Osses, M., 2012. Evaluation of vehicle emission inventories for carbon monoxide and nitrogen oxides for Bogotá, Buenos Aires, Santiago, and São Paulo. Atmos. Environ. 47, 12–19. https://doi.org/10.1016/j.atmosenv.2011.11.051*

*Goldberg, D.L., Anenberg, S.C., Lu, Z., Streets, D.G., Lamsal, L.N., E McDuffie, E., Smith, S.J., 2021. Urban NO x emissions around the world declined faster than anticipated between 2005 and 2019. Environ. Res. Lett. 16, 115004. https://doi.org/10.1088/1748-9326/ac2c34*

*Menares, C., Gallardo, L., Kanakidou, M., Seguel, R., Huneeus, N., 2020. Increasing trends (2001–2018) in photochemical activity and secondary aerosols in Santiago, Chile. Tellus, Ser. B Chem. Phys. Meteorol. 72, 1–18. https://doi.org/10.1080/16000889.2020.1821512*

*\* https://airquality.gsfc.nasa.gov/no2/world/south-and-central-america/santiago*

R2: Comparisons with EDGAR (4): The EDGARv5.0 emission inventory includes estimates of PM emissions from road surface wear and road vehicle tyre and break wear based on the EMEP/EEA guidebook 2019 Tier 1 approach. If I understood correctly, these sources of non-exhaust emissions are not considered in INEMA and could explain some of the discrepancies shown between the two datasets for PM. Please comment on that.

*A: INEMA does not consider non-exhaust PM emissions. The results from external datasets (EDGARv5.0, CAMS, CEDS, LEAP) have been selected only for exhaust emissions from on-road transportation in Chile, assuring the comparison is based on the same source. Attending this comment, we have double-checked this analysis and there is no mixing of exhaust and non-exhaust PM emissions in the comparison.*

R2: Comparisons with EDGAR (5): Figures 7, 8, 9 and 10: Please include the whole time series of the INEMA emissions (up to 2020)

*A: The updated figures, with additional datasets for comparison, include the whole time series for INEMA (1990-2020).*

R2: Effect of COVID-19 restrictions: the time series presented by the authors include the year 2020, which was heavily affected by COVID-19 restrictions. I think it would be very relevant to include a section discussing the results for 2020 and quantifying how they compare to the previous year (2019) (i.e., how total emissions decreased as a consequence of COVID-19). This comment is also linked to the previous one about representing the whole 1990-2020 trend in figures 7 to 10.

*A: We absolutely agree. This emission model was designed and run before COVID-19 effects on mobility and does not consider 2020 reductions in emissions. However, the methodology should incorporate this disruption if the updated official figures of fuel sales are used in the calculation, but the validated 2020 National Energy Balance is not available yet. Nevertheless, there are other recent publications addressing COVID-19 effects on urban vehicle emissions and air quality in Santiago. We have included this issue as a limitation of the dataset (see next answer), offering the reader another reference were COVID-19 impacts have been studied.*

R2: Conclusions: I would recommend to the authors to re-structure the conclusions section and add a new subsection entitled "Limitations of the dataset", in which they clearly state what are the limitations of the current inventory (e.g., non-inclusion of cold-start emissions, use of EU emission factors instead of local EF, ...).

*A: A paragraph on limitations of the dataset has been added to the conclusions.*

R2: Others (1):  Replace MP2.5 for PM2.5 in the text

*A: The acronym was replaced by the English version.*

R2: Others (2): The reference (Gomez, 2020) is missing

*A: The reference was added and the spelling corrected because it should be "Gómez"*

R2: Figure 5: For clarification, I would suggest to change the units to e.g., kg/year. Also, it would be interesting to see the spatial distribution not only of the emissions in specific urban regions but across the whole country.

*A: Since we are building this emission inventory for other users such as ECCAD (https://eccad.aeris-data.fr/) and the information has been uploaded as a doi dataset (http://dx.doi.org/10.17632/z69m8xm843.2), we are using Gg as a common unit for all compounds. For this reason, we consider it is better to present the data in the same format.*

*Regarding regional distribution, it is rather interesting, however the Metropolitan Region dominates with approximately 50% of the national emissions. We considered some approaches for including this analysis, but finally decided not to do it and we would like to maintain this decision.*

R2: Figure 6: Please add a legend

*A: Thanks for the observation, "same as Figure 5" was added to the caption.*

---

## Author Response (AR2)

Author's response

Dear Editor,
Thank you for reviewing our manuscript number ESS-2021-2018. We have gone through all the referee's commentaries and adjusted the manuscript accordingly. After this paragraph you will find our responses. The style used in the response letter is the following: the original general comments made by the editor are kept in normal text (initiated with R), our responses are *in blue italics initiating with A* (Authors).

**Topical Editor (EC1)**
Hugo Denier van der Gon, 05 Jan 2022
Dear Authors,
Thank you for your detailed responses and revision of the MS.
In my opinion the paper is acceptable for publication provided a few minor corrections are made as outlined below:

R1: In your title please change "on-road" to "road" (in your MS_ATC that was not adjusted)

*A: The change has been done and the corrected version uploaded.*

R1: When you introduce the global inventories, please mention that these are not independent. For historic yeas CAMS is mostly based on EDGAR but extrapolated to more recent years using other information such as trends from CEDS.

*A: Thanks for the comment. In page 20, paragraph starting at line 398, we added another reference to CEDS (Smith et al., 2015) and the following phrase:*

*"It is worth mentioning that EDGAR, CAMS and CEDS are not independent. For historic yeas CAMS is mostly based on EDGAR but extrapolated to more recent years, using other information such as trends from CEDS."*

R1: On page 22 bottom of page in MS_ATC version you write "Finally, it is worth noting that CAMS NOx emissions were significantly and lower than EDGAR estimates, although they follow similar trends. This was an unexpected finding, since CAMS emissions of other pollutants were essentially the same as EDGAR's between 1990 and 2011 and differed only in the trends of the following years, becoming linear for CAMS. A revision of CAMS NOx emissions is, therefore, recommended."

*A: Following and accepting next comment, we deleted the phrase mentioned here.*

R1: Please check. I think this is an error in how NOx is reported (as kg NO2 or kg NO). In EDGAR as NO2 and in CAMS as NO. This is a difference of $(14+32) / (14+16) = 1.5333$ - which probably explains the difference you see in Figure 10. If this correct – pleased adjust the above paragraph accordingly.

*A: We appreciate the comment and corrected both the text and Figure 10, using the factor 1.5333. Previously, we detected this difference between NOx reported at the EDGAR official database and EDGAR-ECCAD database but didn't apply the same factor when including CAMS in our analysis.*

R1: Page 23 on your figure 11 you write:
"Considering the differences between EDGAR data and this study's results for Chile, trends in CO/NOx ratio for other European and LAC countries were included."
Please change to for "other European and LAC countries from EDGAR were included" - This is important because you cannot make a similar comparison as for Chile where you have EDGAR and INEMA.

*A: The phrase has been changed according to the suggestion, as follows:*

*"Considering the differences between EDGAR data and this study's results for Chile, trends in CO/NOx ratio for other European and LAC countries from EDGAR were included."*

R1: Furthermore, on figure 11 page 23. "Chile's CO/NOx ratios are in the same range as those found in European countries, which is supported by the fleet renewal shown in Figure 11. Most of the Chilean fleet consists of Euro II/2 and Euro III/3 vehicles, which have much lower CO/NOx ratios that pre-Euro ones. This suggests that vehicle fleet assumptions and, therefore, emission factors used in global emission inventories for LAC countries should be revised."

R1: A few changes in this paragraph are needed.
First sentence: figure 11 does not show fleet renewal but CO/NOx ratio's. The fleet renewal may be the cause for the similarity but it is not shown here. That would be a different figure. Please correct.

*A: The call to Figure 11 has been changed to "Figures 3 and 4", where fleet renewal is shown.*

On the last sentence (emission factors used in global emission inventories for LAC countries should be revised.). I don't think you can generalize to all LAC countries only based on results for Chile as it may not be representative. You suggest it for Chile by showing the local (this study) and the EDGAR data but for the other countries you only show EDGAR ratio's. To conclude more it would be necessary to make a further analysis including national Argentina, Austrian etc. values and compare those with EDGAR, which is beyond the scope of the current paper. Therefore I would suggest to change that sentence to something like "Our analysis for Chile suggests that a careful analysis of national versus global estimates and/or emission factors for road transport emissions is needed for other LAC countries as well.

*A: The suggestion has been made, replacing the phrase " This suggests that vehicle fleet assumptions and, therefore, emission factors used in global emission inventories for LAC countries should be revised" by "Our analysis for Chile suggests that a careful analysis of national versus global estimates and/or emission factors for road transport emissions is needed for other LAC countries as well."*
* * *
*Final comments: We performed a final full revision of the text and found several typos and mistakes when numbering tables, figures, and equations. We made corrections, accordingly, shown on track changes as follows:*
*Page 2, line 51: "For" by "for".*
*Page 2, line 57: "Ministry for the Environment" by "Ministry of the Environment".*
*Page 3, line 73: "Ministry of Environment" by "Ministry of the Environment".*
*Page 7, line 151: "Equation 1" by "Equation 2".*

*Page 7, line 155: "equation 1" by "Equation 2".*
*Page 8, line 159: "Equation 1" by "Equation 2 ".*
*Page 9, line 205: "Table 1" by "Table 4".*
*Page 11, line 241: "section 2.3" by "sections 2.1 and 2.2".*
*Page 13, lines 287 and 290: "Table 6" by "Table 5".*
*Page 14, lines 317, 310, 311 and 315: "Table 6" by "Table 7".*
*Page 14, Table 7: superscript at "-1" for PM.*
*Page 15, line 326: "Table 8" by "Table 7".*
*Page 15, line 328: "Table 8" by "Table 7".*
*Page 15, Table 8: superscript at "-1" for BC and CO.*
*Page 16, Figure 5: "(b) and (c)" by "(c) and (d)" for PM2.5 and BC.*
*Page 17, line 355: "spam" by "span".*
*Page 17, line 371: "2021" by" 2022" for the reference of Alamos et al.*
*Page 22, line 460: "CO (right)" by "NOx (right)".*
*Page 23, line 492: "EDGAR" by "EDGAR, CAMS AND CEDS".*
*Page 25, Acknowledgments: thanks to data providers.*
*Page 25, Financial support: This section has been added, moving funding providers from above section.*
*Page 25, line 554: The first reference (Alamos et al) has been updated.*

---

## Author Response (AR3)

Author's response

Dear Editor,
Thank you for reviewing our manuscript number ESS-2021-2018. We have gone through all the referee's commentaries and adjusted the manuscript accordingly. After this paragraph you will find our responses. The style used in the response letter is the following: the original general comments made by the editor are kept in normal text, our responses are *in blue italics initiating with A* (Authors).

**Comments to the author**:
Dear Authors,
Only two very small technical issues remain
1) page 21, line 5: yeas = years
*A: the correction has been made.*

2) In the acknowledgements: The EDGAR website is fine but for CAMS-GLOB referring to Granier et al. (2019) is enough as the reference is in the reference list, no need for the doi number etc. For ECCAD I suggest to adjust to: "Finally, data provided by the GEIA data portal ECCAD (https://eccad.aeris-data.fr/) has been a great support for this study. "
*A: Both suggestions have been included in the final version.*